# Towards Understanding the Robustness of Diffusion-Based Purification: A Stochastic Perspective

**Yiming Liu**[1][*], **Kezhao Liu**[1][*], **Yao Xiao**[1], **Ziyi Dong**[1], **Xiaogang Xu**[2], **Pengxu Wei**[1,3][†], **Liang Lin**[1,3]
[1]Sun Yat-Sen University, [2]Chinese University of Hong Kong, [3]Peng Cheng Laboratory
{liuym225, liukzh9, xiaoy99, dongzy6}@mail2.sysu.edu.cn
xiaogangxu00@gmail.com, weipx3@mail.sysu.edu.cn, linliang@ieee.org

## Abstract

Diffusion-Based Purification (DBP) has emerged as an effective defense mechanism against adversarial attacks. The success of DBP is often attributed to the forward diffusion process, which reduces the distribution gap between clean and adversarial images by adding Gaussian noise. While this explanation is theoretically sound, the exact role of this mechanism in enhancing robustness remains unclear. In this paper, through empirical analysis, we propose that the intrinsic stochasticity in the DBP process is the primary factor driving robustness. To test this hypothesis, we introduce a novel Deterministic White-Box (DW-box) setting to assess robustness in the absence of stochasticity, and we analyze attack trajectories and loss landscapes. Our results suggest that DBP models primarily rely on stochasticity to avoid effective attack directions, while their ability to purify adversarial perturbations may be limited. To further enhance the robustness of DBP models, we propose Adversarial Denoising Diffusion Training (ADDT), which incorporates classifier-guided adversarial perturbations into the diffusion training process, thereby strengthening the models' ability to purify adversarial perturbations. Additionally, we propose Rank-Based Gaussian Mapping (RBGM) to improve the compatibility of perturbations with diffusion models. Experimental results validate the effectiveness of ADDT. In conclusion, our study suggests that future research on DBP can benefit from a clearer distinction between stochasticity-driven and purification-driven robustness.

## 1 Introduction

Deep learning has achieved remarkable success across various domains, including computer vision (He et al., 2016), natural language processing (OpenAI, 2023), and speech recognition (Radford et al., 2023). However, within this flourishing landscape, the persistent specter of adversarial attacks casts a shadow over the reliability of these neural models. For vision models, adversarial attacks typically involve introducing imperceptible perturbations into input images, tricking the models into producing incorrect outputs with high confidence (Goodfellow et al., 2015; Szegedy et al., 2014). This vulnerability has spurred substantial research into adversarial defenses (Zhang et al., 2019; Samangouei et al., 2018; Shafahi et al., 2019; Wang et al., 2023).

Recently, diffusion-based purification (DBP) (Nie et al., 2022) has emerged as a promising defense against adversarial attacks. Existing studies suggest that DBP robustness primarily stems from the forward diffusion process, which reduces the distribution gap between clean and adversarial images by applying Gaussian noise (Nie et al., 2022; Wang et al., 2022). While this reduction is theoretically supported, its contribution to DBP robustness has not been sufficiently

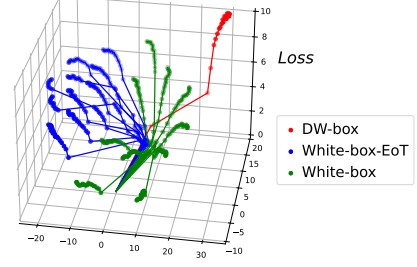

Figure 1: Comparison of attack trajectories under different evaluation settings. The attack trajectory in the standard white-box setting significantly deviates from the DW-box trajectory and demonstrates lower effectiveness.

---

[*]Equal contribution, [†]Corresponding author.

validated through empirical studies. Additionally, experimental results suggest that the stochastic nature of DBP might also play a significant role in enhancing its robustness (Nie et al., 2022).

In this paper, we present a new perspective that emphasizes the role of stochasticity throughout the DBP process as a key contributor to its robustness, challenging the traditional focus on the forward diffusion process. To assess the impact of stochasticity, we propose a Deterministic White-box (DW-box) attack setting, in which the attacker has complete knowledge of both the model parameters and the stochastic elements. Our findings reveal that DBP models experience a significant loss of robustness when the process is made entirely deterministic from the attacker's perspective. Further analysis of attack trajectories and the loss landscape shows that DBP models do not defend against adversarial perturbations by relying on a flat loss landscape, as is common in adversarial training (AT) (Madry et al., 2018); instead, they leverage stochasticity to bypass the most effective attack directions, as illustrated in Figure 1.

Building on this new perspective of DBP robustness, we hypothesize that strengthening the diffusion model's ability to purify adversarial perturbations could further improve the robustness. To test this hypothesis, we propose Adversarial Denoising Diffusion Training (ADDT) for DBP models. This method follows an iterative two-step process: first, the Classifier-Guided Perturbation Optimization (CGPO) step generates adversarial perturbations; then, the diffusion model training step updates the parameters of the diffusion model using these perturbations. To better integrate these perturbations into the diffusion framework, we propose Rank-Based Gaussian Mapping (RBGM), which adjusts the adversarial perturbations to more closely resemble Gaussian noise, in line with the theoretical foundation of diffusion models. Experiments confirm that ADDT consistently enhances the robustness of DBP models. Through further empirical analysis and discussion, we argue that future research on DBP should separate the robustness derived from stochasticity and that achieved through purification. This distinction points to two complementary directions for improving DBP: (1) enhancing its ability to purify adversarial perturbations through efficient training methods, and (2) defending against Expectation over Transformation (EoT) attacks (Athalye et al., 2018b) by increasing the variance of attack gradients.

Our main contributions are as follows:

- We offer a novel perspective on DBP robustness, highlighting the crucial role of stochasticity while challenging the conventional, purification-based view that robustness primarily arises from distribution gap reduction during the forward diffusion process.
- We introduce a new Deterministic White-box attack setting and demonstrate that DBP models rely on stochastic attack gradients to evade the most effective attack directions, revealing key differences from adversarially trained models.
- We show that DBP robustness can be further improved by enhancing the diffusion model's ability to purify adversarial perturbations through the proposed ADDT method.

## 2 RELATED WORK

**Adversarial training (AT).** First introduced by Madry et al. (2018), AT seeks to develop a robust classifier by incorporating adversarial examples into the training process. It has nearly become the de facto standard for improving the adversarial robustness of neural networks (Athalye et al., 2018a; Gowal et al., 2020; Rebuffi et al., 2021). Recent advances in AT harness the generative power of diffusion models to augment training data and prevent overfitting (Gowal et al., 2021; Wang et al., 2023). However, the application of AT to DBP methods has not been thoroughly explored.

**Adversarial purification.** Adversarial purification utilizes generative models to remove adversarial perturbation from inputs before they are processed by downstream models. Traditionally, generative adversarial networks (GANs) (Samangouei et al., 2018) or autoregressive models (Song et al., 2018) have been used as purification models. More recently, diffusion models have been introduced for adversarial purification in a technique known as diffusion-based purification (DBP), which has shown promising results (Nie et al., 2022; Wang et al., 2022; Wu et al., 2022; Xiao et al., 2022). The robustness of DBP models is often attributed to the wash-out effect of Gaussian noise introduced during the forward diffusion process. Nie et al. (2022) propose that the forward diffusion process reduces the Kullback-Leibler (KL) divergence between the distributions of clean and adversarial images. Gao et al. (2022) suggest that while the forward diffusion process improves robustness

by reducing model invariance, the backward process restores this invariance, thereby undermining robustness. However, these theoretical explanations lack strong experimental support.

## 3 PRELIMINARIES

**Adversarial training.** Adversarial training aims to build a robust model by including adversarial samples during training (Madry et al., 2018). This approach can be formulated as a min-max problem, where it first generates adversarial samples (the maximization step) and then adjusts the parameters to resist these adversarial samples (the minimization step). Formally, this can be represented as:

$$\boldsymbol{\theta}^* = \arg\min_{\boldsymbol{\theta}} \mathbb{E}_{(\boldsymbol{x},y)\sim\mathcal{D}} \left[\max_{\boldsymbol{\delta}\in\mathcal{B}} L(f(\boldsymbol{\theta}, \boldsymbol{x}+\boldsymbol{\delta}), y)\right], \tag{1}$$

where $L$ is the loss function, $f$ is the classifier, $(\boldsymbol{x}, y) \sim \mathcal{D}$ denotes the sampling of training data from the distribution $\mathcal{D}$, and $\mathcal{B}$ defines the set of permissible perturbation $\boldsymbol{\delta}$.

**Diffusion models.** Denoising Diffusion Probabilistic Models (DDPM) (Ho et al., 2020) and Denoising Diffusion Implicit Models (DDIM) (Song et al., 2020) simulate a gradual transformation in which noise is progressively added to an image and then removed to restore the original image. The forward process can be represented as:

$$\boldsymbol{x}_t = \sqrt{\overline{\alpha}_t}\boldsymbol{x}_0 + \sqrt{1-\overline{\alpha}_t}\boldsymbol{\epsilon}, \quad \boldsymbol{\epsilon} \sim \mathcal{N}(\mathbf{0}, \boldsymbol{I}), \tag{2}$$

where $\boldsymbol{x}_0$ is the original image and $\boldsymbol{x}_t$ is the noisy image. $\overline{\alpha}_t$ is the cumulative noise level at step $t$ ($1 < t \leq T$, where $T$ is the number of diffusion training steps). The parameters $\boldsymbol{\theta}$ are optimized by minimizing the distance between the actual and estimated noise:

$$\boldsymbol{\theta}^* = \arg\min_{\boldsymbol{\theta}} \mathbb{E}_{\boldsymbol{x}_0, t, \boldsymbol{\epsilon}} \left[\|\boldsymbol{\epsilon} - \boldsymbol{\epsilon}_{\boldsymbol{\theta}}(\sqrt{\overline{\alpha}_t}\boldsymbol{x}_0 + \sqrt{1-\overline{\alpha}_t}\boldsymbol{\epsilon}, t)\|_2^2\right], \tag{3}$$

where $\boldsymbol{\theta}^*$ represents the optimized parameters, and $\boldsymbol{\epsilon}_{\boldsymbol{\theta}}$ is the model's predicted noise. Using $\boldsymbol{\epsilon}_{\boldsymbol{\theta}^*}$, we can estimate $\hat{\boldsymbol{x}}_0$ in a single step:

$$\hat{\boldsymbol{x}}_0 = \left(\boldsymbol{x}_t - \sqrt{1-\overline{\alpha}_t}\boldsymbol{\epsilon}_{\boldsymbol{\theta}^*}(\boldsymbol{x}_t, t)\right)/\sqrt{\overline{\alpha}_t}, \tag{4}$$

where $\hat{\boldsymbol{x}}_0$ is the recovered image. DDPM typically takes an iterative approach to restore the image, removing a small amount of Gaussian noise at a time:

$$\hat{\boldsymbol{x}}_{t-1} = \left(\boldsymbol{x}_t - \frac{\beta_t}{\sqrt{1-\overline{\alpha}_t}}\boldsymbol{\epsilon}_{\boldsymbol{\theta}^*}(\boldsymbol{x}_t, t)\right)/\sqrt{1-\beta_t} + \sqrt{\beta_t}\boldsymbol{\epsilon}, \tag{5}$$

where $\beta_t$ is the noise level at step $t$, $\hat{\boldsymbol{x}}_{t-1}$ is the image recovered at step $t-1$, and $\boldsymbol{\epsilon}$ is sampled from $\mathcal{N}(\mathbf{0}, \boldsymbol{I})$. DDIM speeds up the denoising process by skipping certain intermediate steps. Recent work suggests that DDPM could also benefit from a similar approach (Nichol & Dhariwal, 2021). Score SDEs (Song et al., 2021a) provide a score function perspective on DDPM and further lead to the derivations of DDPM++ (VPSDE) and EDM (Karras et al., 2022). In this diffusion process, the noise terms $\boldsymbol{\epsilon}$ in Equation (2) and Equation (5) represent the key stochastic elements that control the randomness of the process. More discussion of stochastic elements is provided in Appendix C.1.

**Diffusion-based purification (DBP).** DBP uses diffusion models to remove adversarial perturbation from images. Instead of using a complete diffusion process between the clean image and pure Gaussian noise (between $t = 0$ and $t = T$), they first diffuse $\boldsymbol{x}_0$ to a predefined timestep $t = t^*(t^* < T)$ via Equation (2), and recover the image $\hat{\boldsymbol{x}}_0$ via the reverse diffusion process in Equation (5).

## 4 STOCHASTICITY-DRIVEN ROBUSTNESS OF DBP

### 4.1 STOCHASTICITY AS THE MAIN FACTOR OF DBP ROBUSTNESS

As discussed in Section 2, previous studies primarily attribute the robustness of DBP models to the forward diffusion process, which perturbs inputs with Gaussian noise to reduce the distribution gap between adversarial and clean images (Nie et al., 2022; Wang et al., 2022). As a result, adversarial perturbations can be "washed out" by the Gaussian noise. However, it has also been observed that the robustness of DiffPure diminishes when switching from Stochastic Differential Equation (SDE) sampling to Ordinary Differential Equation (ODE) (Nie et al., 2022), which introduces less stochasticity. This reduction in robustness cannot be fully explained by the "wash out" effect of Gaussian noise, suggesting that stochasticity plays a role in DBP robustness.

To assess the impact of stochasticity on DBP robustness, we implemented both DDPM and DDIM within the DiffPure framework (Nie et al., 2022) and compared their performance (referred to as **DP$_{DDPM}$** and **DP$_{DDIM}$**). Note that the original implementation of DiffPure adopts a DDPM discretization form of DDPM++ (VPSDE), which differs only minimally from DDPM (Ho et al., 2020). Thus, the primary difference between DiffPure and our DP$_{DDPM}$ is that DiffPure employs a larger UNet. DDPM utilizes a stochastic SDE-based reverse process, which introduces Gaussian noise in both the forward and reverse processes, making the entire process stochastic. In contrast, DDIM uses a deterministic ODE-based reverse process and introduces Gaussian noise only in the forward process. The results are shown in Figure 2 (labeled as *Clean* and *White*), where *White* refers to $\ell_\infty$ white-box PGD (Madry et al., 2018) + EoT (Athalye et al., 2018b) attacks (as detailed in Section 6.1) Although DP$_{DDPM}$ achieves higher clean accuracy, it shows lower robust accuracy under adaptive white-box attacks, consistent with the findings of Nie et al. (2022). This suggests that the stochasticity in the reverse diffusion process may also play a crucial role in DBP robustness.

To better control the stochasticity in both the forward and reverse diffusion processes and to isolate its effect on DBP robustness, we introduce a novel attack setting called the **Deterministic White-Box** (DW-box) setting. In this setting, the attacker not only has complete knowledge of the model parameters but also of the exact values of the stochastic elements sampled during evaluation, effectively making the diffusion process deterministic from the attacker's perspective. This setting could be realistic if the attacker knows the seed or the initial random state used for pseudo-random number generation in the model. For our evaluations, we define three levels of attacker knowledge: (1) Conventional **White-box** setting, where the attacker has access to the model parameters but not the stochastic elements; (2) **DW$_{Fwd}$-box/DW$_{Rev}$-box** setting, where the attacker knows the stochastic elements in the forward/reverse process in addition to the model parameters; (3) **DW$_{Both}$-box** setting, where the attacker has complete knowledge of the model parameters and all stochastic elements in both the forward and reverse processes. Details of these settings are provided in Appendix C.2.

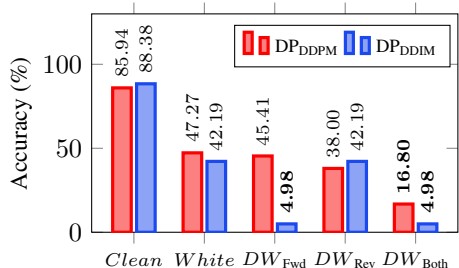

Figure 2: DP$_{DDPM}$ and DP$_{DDIM}$ robust accuracy under different attack settings on CIFAR-10. Both models lose most of their robustness only when the attacker knows all stochastic elements (highlighted in bold: DW$_{Both}$-box for DP$_{DDPM}$/DP$_{DDIM}$ and DW$_{Fwd}$-box for DP$_{DDIM}$).

With the **Deterministic White-Box** attack, we are able to compare traditional theories with our proposed hypothesis. These explanations diverge in behavior when stochasticity is controlled. Traditional theories emphasize the forward diffusion process as the primary defense mechanism, suggesting that both DP$_{DDPM}$ and DP$_{DDIM}$ should behave similarly under the DW$_{Fwd}$-box setting. In contrast, our hypothesis emphasizes the stochasticity throughout the diffusion process as the crucial factor. We hypothesize that as DP$_{DDIM}$ becomes deterministic under the DW$_{Fwd}$-box setting, it should experience a significant reduction in robustness—similar to DP$_{DDPM}$ under the DW$_{Both}$-box setting. We evaluated adversarial robustness on CIFAR-10 using $\ell_\infty$ attacks (see Section 6.1). The results are shown in Figure 2. Under the DW$_{Fwd}$-box setting, DP$_{DDPM}$ maintains a significant portion of its robustness, whereas DP$_{DDIM}$ loses almost all resistance to adversarial attacks. This phenomenon can not be explained by the "wash out" theory. Furthermore, under the DW$_{Both}$-box setting, DP$_{DDPM}$ shows a substantial drop similar to DP$_{DDIM}$ in the DW$_{Fwd}$ setting. This suggests that stochasticity in both the forward and reverse diffusion processes plays a critical role in maintaining robustness.

Our findings suggest that DBP models primarily rely on stochasticity to resist adversarial attacks, rather than depending solely on the forward diffusion process, and it also reveals that DBP models lack the ability to *effectively purify adversarial perturbations*.

### 4.2 EXPLAINING STOCHASTICITY-DRIVEN ROBUSTNESS

When attacking stochastic models, a commonly used technique is EoT (Athalye et al., 2018b). To assess the influence of adaptive attacks, we compare the performance of white-box attacks with and without using EoT. The results in Table 1 show that EoT-based adaptive attacks have only a modest impact on robustness, which contrasts sharply with DW-box attacks. A more detailed discussion of EoT steps is provided in Appendix A.

To compare different attacks, we visualize the attack trajectories using t-SNE. These trajectories are projected onto the *xy*-plane, with loss values plotted on the *z*-axis. We compare three types of attacks: white-box without EoT (White-box), white-box with EoT (White-box-EoT), and Deterministic White-box (DW-box). As shown in Figure 1, the trajectories vary significantly across all settings, re-

Table 1: Evaluation of DBP methods under various attack settings shows that EoT significantly affects the model's robustness accuracy (%).

| Metric | DiffPure | GDMP (MSE) | $DP_{DDPM}$ | $DP_{DDIM}$ |
|---|---|---|---|---|
| Clean | 89.26 | 91.80 | 85.94 | 88.38 |
| PGD20 | 69.04 | 53.13 | 60.25 | 54.59 |
| PGD20+EoT10 | 55.96 | 40.97 | 47.27 | 42.19 |

flecting the stochasticity of DBP models. Notably, DW-box attacks lead to a significant increase in loss values, whereas white-box attacks—even with EoT—result in only moderate increases. This observation suggests that *stochasticity prevents attackers from finding the optimal attack direction*. Even if the EoT estimate accurately captures the mean gradient direction, the high variance in attack gradients may prevent alignment with the optimal attack direction (the direction of the DW-box attack). This leads to reduced attack performance. Additional evidence is provided in Appendix B.

To study how different attacks impact robust accuracy, we compared the loss landscapes under White-box-EoT and Deterministic White-box attacks. We computed the average loss variation across a batch of images and present the results in Figure 3. The trajectory of the White-box-EoT attack deviates from that of the Deterministic White-box attack, resulting in a flatter loss landscape along its path, whereas the Deterministic White-box attack produces a steep increase in loss. This observation suggests that the inherent stochasticity in DBP models prevents White-box-EoT attacks from identifying the optimal attack direction, and that removing this stochasticity makes the model vulnerable to adversarial perturbations. These findings contrast with adversarially trained models, where the loss landscape remains flat even along adver-

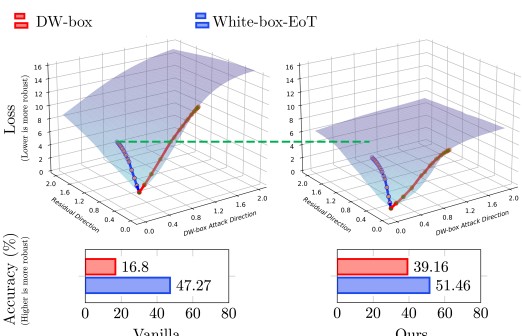

Figure 3: Visualisation of attack trajectories for White-box-EoT attacks and DW-box attacks on the loss landscape. The loss landscape is steep in the direction of the DW-box attack. The plot is based on the first 128 images of CIFAR-10.

sarial directions (Shafahi et al., 2019). Further details on the loss landscape visualization can be found in Appendix F.

In conclusion, we propose that DBP models, instead of *exhibiting a flat loss landscape*, leverage stochasticity to *evade the most effective attack directions*. Note that while certified defense methods such as random smoothing also incorporate stochasticity (Xiao et al., 2022; Carlini et al., 2022), their mechanisms and implications differ from those of DBP methods (see Appendix D).

## 5 TOWARDS IMPROVING THE PURIFICATION CAPABILITY OF DBP MODELS

Based on the analysis in Section 4, and considering the loss increase along the DW-box attack direction, we propose that as an alternative to stochasticity-driven robustness, the performance of DBP can be further improved by flattening the loss landscape. Achieving this requires incorporating adversarial samples into the training of DBP models. From the perspective of adversarial purification, this involves enhancing the diffusion model's ability to purify adversarial perturbations.

To address this, we propose **Adversarial Denoising Diffusion Training (ADDT)**, a method that incorporates adversarial perturbations into the training of diffusion models in DBP. ADDT follows an iterative two-step process: (1) **Classifier-Guided Perturbation Optimization (CGPO)**, which generates adversarial perturbations by maximizing the classification error of a pre-trained classifier; (2) **Diffusion Model Training**, which trains the diffusion model on these perturbations to improve its ability to purify adversarial perturbations.

Integrating adversarial perturbations into diffusion training is challenging due to the Gaussian noise assumption inherent in diffusion models. To overcome this, we propose **Rank-Based Gaussian Mapping (RBGM)**, a technique that transforms adversarial perturbations into a form more consistent

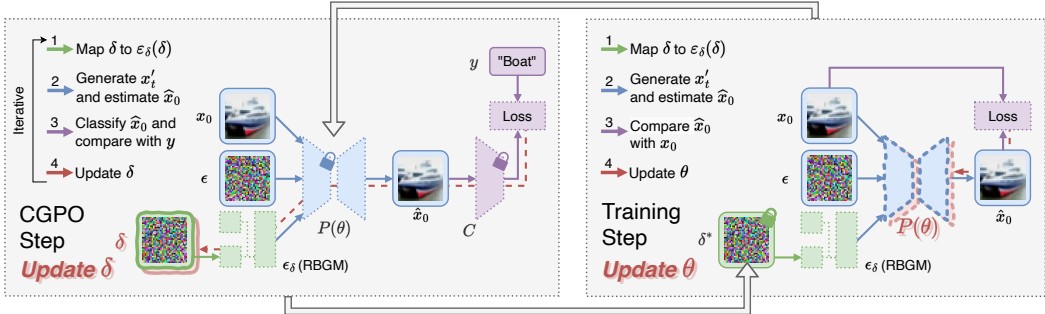

Figure 4: Overview of Adversarial Denoising Diffusion Training (ADDT). ADDT alternates between a CGPO step (left grey box) to refine the perturbations with a frozen diffusion model and classifier, and a training step (right grey box) to update the diffusion model with the refined perturbation. Throughout the process, RBGM is used to make the perturbation more "Gaussian-like".

with the Gaussian noise assumption, thereby facilitating their integration into the diffusion training process.

Figure 4 provides an overview of ADDT, and pseudocode is available in Appendix G. The following subsections detail the components of ADDT.

## 5.1 ADVERSARIAL DENOISING DIFFUSION TRAINING

**Classifier-Guided Perturbation Optimization (CGPO) step.** In this step, we refine the adversarial perturbations $\boldsymbol{\delta}$ to maximize the classification error of a pre-trained classifier $C$, as shown on the left side of Figure 4. The process begins by reconstructing a clean image $\hat{\boldsymbol{x}}_0$ from the perturbed input $\boldsymbol{x}'_t$ using the diffusion model $\boldsymbol{P}$. The perturbed input $\boldsymbol{x}'_t$ is generated from the original image $\boldsymbol{x}_0$, Gaussian noise $\boldsymbol{\epsilon}$, and the RBGM-mapped perturbation $\boldsymbol{\epsilon}_{\boldsymbol{\delta}}(\boldsymbol{\delta})$, as detailed in Section 5.2. Here, $\boldsymbol{P}(\boldsymbol{\theta}, \boldsymbol{x}'_t, t)$ represents a one-step diffusion process based on the formulation in Equation (4). It reconstructs the image $\hat{\boldsymbol{x}}_0$ from the noisy input $\boldsymbol{x}'_t$. The classifier $C$ is then used to predict a label for this reconstructed image $\hat{\boldsymbol{x}}_0$. The optimization objective is to maximize the discrepancy between the classifier's prediction and the ground-truth label $y$. This can be formulated as:

$$\boldsymbol{\delta}^* = \arg\max_{\boldsymbol{\delta}} \mathbb{E}_{\boldsymbol{x}_0, t, \boldsymbol{\epsilon}} \left[ L\left( C\left( \boldsymbol{P}\left(\boldsymbol{\theta}, \boldsymbol{x}'_t(\boldsymbol{x}_0, \boldsymbol{\epsilon}, \boldsymbol{\epsilon}_{\boldsymbol{\delta}}(\boldsymbol{\delta})), t\right)\right), y\right) \right], \tag{6}$$

where $L(\cdot, y)$ denotes the loss function measuring the discrepancy between the classifier's prediction and the ground-truth label $y$. The pre-trained classifier $C$ provides semantic guidance and need not be identical to the protected classifier, as detailed in Section 6.2. To optimize $\boldsymbol{\delta}$, we employ the PGD attack (Madry et al., 2018). Since RBGM is non-differentiable, we apply the Backward Pass Differentiable Approximation (BPDA) approach (Athalye et al., 2018a). By applying BPDA, we employ RBGM during forward propagation but bypass it during backpropagation. Note that BPDA/RBGM is only applied during training and does not affect the evaluation of DBP robustness.

**Diffusion Model Training step.** The objective of this step is to update the diffusion model's parameters so that it can reconstruct the original image $\boldsymbol{x}_0$ from its perturbed counterpart $\boldsymbol{x}'_t$. As illustrated on the right side of Figure 4, the model aims to remove both Gaussian noise and adversarial perturbations, effectively denoising the input. The optimization objective is defined as:

$$\boldsymbol{\theta}^* = \arg\min_{\boldsymbol{\theta}} \mathbb{E}_{\boldsymbol{x}_0, t, \boldsymbol{\epsilon}} \left[ \frac{\sqrt{\overline{\alpha}_t}}{\sqrt{1 - \overline{\alpha}_t}} \left\| \boldsymbol{x}_0 - \boldsymbol{P}(\boldsymbol{\theta}, \boldsymbol{x}'_t(\boldsymbol{x}_0, \boldsymbol{\epsilon}, \boldsymbol{\epsilon}_{\boldsymbol{\delta}}(\boldsymbol{\delta})), t) \right\|_2^2 \right], \tag{7}$$

where the scaling factor $\sqrt{\overline{\alpha}_t}/\sqrt{1 - \overline{\alpha}_t}$ ensures consistency with the standard DDPM/DDIM loss formulation.

## 5.2 RANK-BASED GAUSSIAN MAPPING

Traditional diffusion models assume that input images are corrupted by independent Gaussian noise $\boldsymbol{\epsilon}$. To preserve Gaussian-like characteristics while encoding adversarial features, we introduce Rank-Based Gaussian Mapping (RBGM), as shown in Figure 5. RBGM, denoted as $\boldsymbol{\epsilon}_{\boldsymbol{\delta}}(\boldsymbol{\delta})$, retains the

ordering of the elements in the input tensor $\boldsymbol{\delta}$ while replacing their original values with samples drawn from a standard Gaussian distribution $\boldsymbol{\epsilon}_s$. Specifically, we first sample a Gaussian tensor $\boldsymbol{\epsilon}_s$ with dimensions matching $\boldsymbol{\delta}$. Next, we sort the elements of both $\boldsymbol{\delta}$ and $\boldsymbol{\epsilon}_s$ in ascending order. By mapping the sorted elements of $\boldsymbol{\delta}$ to the corresponding elements of $\boldsymbol{\epsilon}_s$, we obtain $\boldsymbol{\epsilon}_{\boldsymbol{\delta}}(\boldsymbol{\delta})$, which approximates a Gaussian distribution while preserving the structural information in $\boldsymbol{\delta}$. To further refine the perturbation towards Gaussianity, we blend the RBGM-mapped perturbation with additional randomly sampled Gaussian noise. The adversarial input $\boldsymbol{x}'_t$ is then defined as follows:

$$\boldsymbol{x}'_t(\boldsymbol{x}_0, \boldsymbol{\epsilon}, \boldsymbol{\epsilon}_{\boldsymbol{\delta}}(\boldsymbol{\delta})) = \sqrt{\overline{\alpha_t}}\boldsymbol{x}_0 + \sqrt{1 - \lambda_t^2}\sqrt{1 - \overline{\alpha_t}}\boldsymbol{\epsilon} + \lambda_t\sqrt{1 - \overline{\alpha_t}}\boldsymbol{\epsilon}_{\boldsymbol{\delta}}(\boldsymbol{\delta}),  \tag{8}$$

where $\lambda_t$ modulates the level of adversarial perturbation. This ensures that the overall perturbation remains largely independent of $\boldsymbol{x}_0$, preventing the perturbations from overwhelming the denoising model's learning process. We determine $\lambda_t$ using the following formulation:

$$\lambda_t = \texttt{clip}(\gamma_t \lambda_{unit}, \lambda_{min}, \lambda_{max}), \quad \gamma_t = \sqrt{\overline{\alpha_t}}/\sqrt{1 - \overline{\alpha_t}},  \tag{9}$$

where the $\texttt{clip}$ function constrains $\lambda_t$ between $\lambda_{min}$ and $\lambda_{max}$. Additional details and discussions about RBGM can be found in Appendix K.

## 6 EXPERIMENTS AND DISCUSSIONS

### 6.1 EXPERIMENT SETUP

**Classifier.** We train a WideResNet-28-10 model for 200 epochs following the methods in Yoon et al. (2021); Wang et al. (2022), achieving an accuracy of 95.12% on CIFAR-10 and 76.66% on CIFAR-100 dataset.

**DBP timestep.** For the diffusion forward process, we adopt the same timestep settings as DiffPure (Nie et al., 2022). In continuous-time models, such as the VPSDE (DDPM++) variant, with the forward time parameter $0 \le t \le 1$, we set $t^* = 0.1$, balancing noise introduction and computational efficiency. For discrete-time models, such as DDPM and DDIM, where $t = 0, 1, ..., T$, we similarly set the timestep to $t^* = 0.1 \times T$. Additional settings and results for $DP_{EDM}$ are provided in Appendix I.

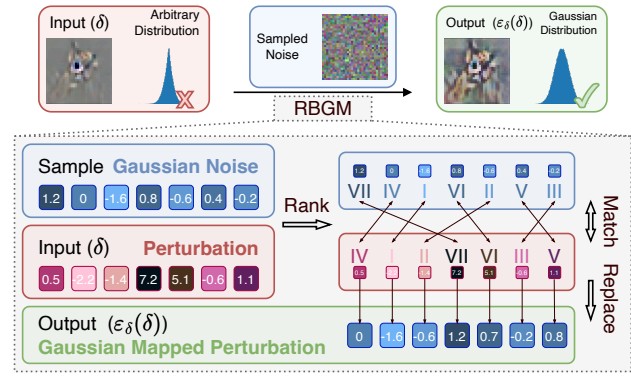

Figure 5: Rank-Based Gaussian Mapping. RBGM trims the input to follow Gaussian distribution. It samples a Gaussian noise and then replaces elements in the input with corresponding values from the noise, based on matching ranks.

**Robustness evaluation.** We assess model robustness using the PGD20+EoT10 attack (Athalye et al., 2018b). For $\ell_\infty$-norm attacks, we set the step size $\alpha = 2/255$ and the maximum perturbation $\epsilon = 8/255$; for $\ell_2$-norm attacks, we use $\alpha = 0.1$ and $\epsilon = 0.5$. Due to the high computational cost of EoT attacks, we evaluate the models on the first 1024 images from the CIFAR-10 and CIFAR-100 datasets. For comprehensive evaluation settings, including an analysis of AutoAttack (Croce & Hein, 2020) performance on DBP models, refer to Appendix E.

Table 2: Clean and robust accuracy (%) on CIFAR-10, obtained by different classifiers. ADDT (WRN-28-10 guidance) improves robustness in protecting different subsequent classifiers. (*: the classifier used in ADDT fine-tuning).

| DBP method | Classifier | Vanilla | | | ADDT | | |
|---|---|---|---|---|---|---|---|
| | | Clean | $\ell_\infty$ | $\ell_2$ | Clean | $\ell_\infty$ | $\ell_2$ |
| $DP_{DDPM}$ | VGG-16 (Simonyan & Zisserman, 2014) | 84.77 | 41.99 | 66.89 | 85.06 | **46.09** | **67.87** |
| | ResNet-50 (He et al., 2016) | 83.11 | 44.04 | 67.58 | 83.84 | **48.14** | **67.87** |
| | WRN-28-10* (Zagoruyko & Komodakis, 2016) | 85.94 | 47.27 | 69.34 | 85.64 | **51.46** | **70.12** |
| | WRN-70-16 (Zagoruyko & Komodakis, 2016) | 88.43 | 48.93 | 70.31 | 87.84 | **52.54** | **70.70** |
| | ViT-B (Dosovitskiy et al., 2020) | 85.45 | 45.61 | 69.53 | 85.25 | **48.63** | **69.92** |
| $DP_{DDIM}$ | VGG-16 (Simonyan & Zisserman, 2014) | 87.16 | 29.00 | 61.82 | 87.55 | **35.06** | **66.11** |
| | ResNet-50 (He et al., 2016) | 86.04 | 31.74 | 62.11 | 86.57 | **38.77** | **65.82** |
| | WRN-28-10* (Zagoruyko & Komodakis, 2016) | 88.96 | 43.16 | 67.58 | 88.18 | **47.85** | **70.61** |
| | WRN-70-16 (Zagoruyko & Komodakis, 2016) | 84.40 | 39.16 | 68.36 | 84.96 | **47.66** | **69.14** |
| | ViT-B (Dosovitskiy et al., 2020) | 88.77 | 34.38 | 65.72 | 88.48 | **41.02** | **68.65** |
| $DP_{EDM}$ | WRN-28-10* (Zagoruyko & Komodakis, 2016) | 86.43 | 62.50 | 76.86 | 86.33 | **66.41** | **79.16** |
| | WRN-70-16 (Zagoruyko & Komodakis, 2016) | 86.62 | 65.62 | 76.46 | 86.43 | **69.63** | 78.91 |

Table 3: Clean and robust accuracy (%) on CIFAR-10 obtained by different DBP methods. All methods show consistent improvement fine-tuned with ADDT.

| Diffusion model | DBP method | Clean | $\ell_\infty$ | $\ell_2$ |
|---|---|---|---|---|
| - | - | 95.12 | 0.00 | 1.46 |
| DDIM | $DP_{DDIM}$ | 88.38 | 42.19 | 70.02 |
| | **$DP_{DDIM}$+ADDT** | 88.77 | **46.48** | **71.19** |
| DDPM | GDMP (No Guided) (Wang et al., 2022) | 91.41 | 40.82 | 69.63 |
| | GDMP (MSE) (Wang et al., 2022) | 91.80 | 40.97 | 70.02 |
| | GDMP (SSIM) (Wang et al., 2022) | 92.19 | 38.18 | 68.95 |
| | $DP_{DDPM}$ | 85.94 | 47.27 | 69.34 |
| | **$DP_{DDPM}$+ADDT** | 85.64 | **51.46** | **70.12** |
| DDPM++ | COUP (Zhang et al., 2024) | 90.33 | 50.78 | 71.19 |
| | DiffPure | 89.26 | 55.96 | 75.78 |
| | **DiffPure+ADDT** | 89.94 | **62.11** | **76.66** |
| EDM | $DP_{EDM}$ (Appendix I) | 86.43 | 62.50 | 76.86 |
| | **$DP_{EDM}$+ADDT** (Appendix I) | 86.33 | **66.41** | **79.16** |

Table 4: Clean and robust accuracy (%) on $DP_{DDPM}$. ADDT improves robustness across different NFEs, especially at lower NFEs (*: default DDPM generation setting; -: classifier only).

| Dataset | NFEs | Vanilla | | | ADDT | | |
|---|---|---|---|---|---|---|---|
| | | Clean | $\ell_\infty$ | $\ell_2$ | Clean | $\ell_\infty$ | $\ell_2$ |
| CIFAR-10 | - | 95.12 | 0.00 | 1.46 | 95.12 | 0.00 | 1.46 |
| | 5 | 49.51 | 21.78 | 36.13 | 59.96 | **30.27** | **41.99** |
| | 10 | 73.34 | 36.72 | 55.47 | 78.91 | **43.07** | **62.97** |
| | 20 | 81.45 | 45.21 | 65.23 | 83.89 | **48.44** | **69.82** |
| | 50 | 85.54 | 46.78 | 68.85 | 85.45 | **50.20** | **69.04** |
| | 100* | 85.94 | 47.27 | 69.34 | 85.64 | **51.46** | **70.12** |
| CIFAR-100 | - | 76.66 | 0.00 | 2.44 | 76.66 | 0.00 | 2.44 |
| | 5 | 17.29 | 3.71 | 9.28 | 21.78 | **6.25** | **13.77** |
| | 10 | 34.08 | 10.55 | 19.24 | 40.62 | **14.55** | **27.25** |
| | 20 | 48.05 | 17.68 | 30.66 | 53.32 | **18.65** | **36.13** |
| | 50 | 55.57 | 20.02 | 37.70 | 59.47 | **22.75** | **40.72** |
| | 100* | 57.52 | 20.41 | 37.89 | 59.18 | **23.73** | **41.70** |

**ADDT.** ADDT fine-tuning is guided by the pre-trained WideResNet-28-10 classifier. For the CIFAR-10 dataset, we utilize the pre-trained exponential moving average (EMA) diffusion model developed by Ho et al. (2020) (converted to Huggingface Diffusers format by Fang et al. (2023)). For the CIFAR-100 dataset, we fine-tune the CIFAR-10 diffusion model for 100 epochs. In CGPO, we set the hyperparameters to $\lambda_{unit} = 0.03$, $\lambda_{min} = 0$, and $\lambda_{max} = 0.3$, and iteratively refine the perturbation $\delta$ for 5 steps. Additional details regarding computational cost are provided in Appendix O.

## 6.2 DEFENSE PERFORMANCE UNDER DIFFERENT CONDITIONS

**Performance across different DBP models.** We evaluate the effectiveness of ADDT by applying it to various diffusion models and comparing their performance with existing methods. The results shown in Table 3 indicate that ADDT consistently enhances model robustness while maintaining competitive accuracy on clean data.

**Performance across different downstream classifiers.** We evaluate ADDT's cross-classifier protection performance by applying a diffusion model specifically ADDT fine-tuned with WRN-28-10 guidance, to protect a variety of classifiers. The results in Table 2 show that these models effectively protect classifiers with diverse architectures. Notably, employing a $DP_{EDM}$ model trained with WRN-28-10 guidance achieves a $69.63\%$ $\ell_\infty$ robust accuracy on a WRN-70-16 classifier. This ability to protect different classifiers without classifier-specific fine-tuning highlights the effectiveness and feasibility of ADDT.

Table 5: Clean and robust accuracy (%) on CIFAR-10 fine-tuned with different training samples. (None: no fine-tuning)

| DBP method | Training samples | Clean | $\ell_\infty$ | $\ell_2$ |
|---|---|---|---|---|
| $DP_{DDPM}$ | None | 85.94 | 47.27 | 69.34 |
| | Clean | 85.25 | 47.27 | 68.26 |
| | MSE-guided | 86.91 | 46.97 | **70.80** |
| | CGPO | 85.64 | **51.46** | 70.12 |
| $DP_{DDIM}$ | None | 88.96 | 43.16 | 67.58 |
| | Clean | 88.87 | 41.41 | 67.19 |
| | MSE-guided | 89.36 | 40.92 | 67.68 |
| | CGPO | 88.18 | **47.85** | **70.61** |

**Performance under acceleration.** Speeding up the diffusion process by omitting intermediate steps has become a common practice in diffusion models (Song et al., 2020; Nichol & Dhariwal, 2021). In this context, we evaluate the robustness of accelerated DBP models. To quantify computational cost, we utilize the number of Neural Function Evaluations (NFEs), which reflects the number of evaluation steps executed during the DBP reverse process. For our experiments, we set $t^* = 0.1 \times T$ and accelerate the process by excluding intermediate time steps. For example, with 5 NFEs, the time steps for the DBP reverse process would be $t = [100, 80, 60, 40, 20, 0]$. The results in Table 4 validate the effectiveness of ADDT in improving the robustness of accelerated $DP_{DDPM}$ models. Note that the performance of $DP_{DDPM}$ varies significantly with different NFE values. This may be because DDPM introduces stochasticity (Gaussian noise) at each reverse step, and with fewer reverse steps, the influence of this stochasticity diminishes. Additionally, the generative performance of DDPM is sensitive to the omission of intermediate steps. We also evaluated $DP_{DDIM}$ models, as detailed in Appendix H.

## 6.3 ABLATION STUDY AND ANALYSIS

**RBGM.** We compare the generative ability of diffusion models fine-tuned from the same pre-trained

models, using two different perturbations: RBGM-mapped perturbations and $\ell_\infty$ perturbations. To quantify generation quality, we use Fréchet Inception Distance (FID) scores (Heusel et al., 2017), as shown in Table 6. The results demonstrate that diffusion models fine-tuned with RBGM-mapped perturbations maintain generation quality comparable to the vanilla diffusion model, while models directly fine-tuned with $\ell_\infty$ perturbations without RBGM show degraded

Table 6: FID score of DDPM fine-tuned on CIFAR-10 with different perturbations (lower is better). Fine-tuning with RBGM-mapped perturbations results in lower FID scores compared to $\ell_\infty$ perturbations (without RBGM).

| | Vanilla | Clean Fine-tuning | ADDT | ADDT w/o RBGM |
|---|---|---|---|---|
| FID | 3.196 | 3.500 | 5.190 | 13.608 |

performance. Additionally, we observe that training with RBGM-mapped perturbations generalizes better to different types of attacks. Experimental details and further tests are provided in Appendix L.

**CGPO.** We analyze the impact of fine-tuning with different training samples in Table 5. Specifically, we compare the performance of samples generated with classifier guidance in the CGPO step, referred to as "CGPO", against those generated with Mean Squared Error (MSE) loss, denoted as "MSE-guided". The evaluation results are presented for DDPM with 100 NFEs and DDIM with 10 NFEs. The results demonstrate that samples generated by CGPO significantly outperform MSE-guided samples in terms of enhancing DBP robustness.

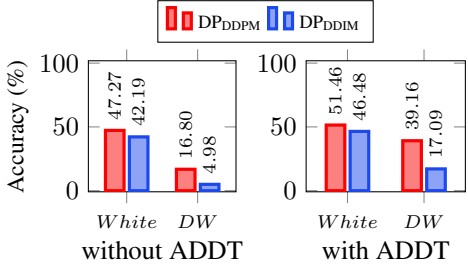

Figure 6: Revisiting robustness under Deterministic White-box setting. ADDT improves robustness under both White-box and Deterministic White-box settings, implying that ADDT strengthens the models' ability to purify adversarial inputs.

**Revisiting DBP robustness.** We re-examine robustness under the Deterministic White-box setting by comparing the performance of diffusion models with and without ADDT fine-tuning, as shown in Figure 6. The fine-tuned models show significantly higher robust accuracy under the DW-box setting, indicating that ADDT improves non-stochasticity-based robustness. Further experiments across different models and NFEs, presented in Appendix M, confirm these improvements in robustness. We also compare the loss landscapes of ADDT fine-tuned models and vanilla diffusion models, as shown in Figure 3. This comparison reveals that our method effectively smooths the loss landscape of DBP models, enhancing their ability to resist adversarial perturbations.

**Evaluation with stronger PGD+EoT attacks.** To balance computational cost and attack strength, we primarily employ the PGD20+EoT10 configuration in our evaluations. To further validate the efficacy of ADDT under stronger attack settings, we assess its performance using the more challenging PGD200+EoT20 setup. The results presented in Table 7 and Table 9 show that under these intensified attacks, ADDT's robust accuracy experiences a moderate 5% drop compared to the PGD20+EoT10 setting. Nevertheless, across various configurations and datasets, ADDT consistently outperforms the baseline in terms of robust accuracy.

## 6.4 SCALING TO MORE COMPLEX AND HIGH-DIMENSIONAL DATA

To assess the scalability of DBP and ADDT on more complex datasets, we extend our experiments to include Tiny-ImageNet (Le & Yang, 2015) and ImageNet-1k (Deng et al., 2009). For Tiny-ImageNet, we trained the diffusion model from scratch for 200 epochs and then fine-tuned it with

Table 7: Robust accuracy (%) on CIFAR-10 under stronger attacks.

| DBP method | PGD200+EoT20 | | PGD20+EoT10 | |
|---|---|---|---|---|
| | Vanilla ($\ell_\infty$) | ADDT ($\ell_\infty$) | Vanilla ($\ell_\infty$) | ADDT ($\ell_\infty$) |
| $DP_{DDPM}$ | 41.02 | **46.19** | 47.27 | **51.46** |
| $DP_{DDIM}$ | 36.23 | **41.11** | 43.16 | **47.85** |
| DiffPure | 48.93 | **55.76** | 55.96 | **62.11** |

ADDT for 50 epochs, using a pretrained WRN-28-10 classifier for guidance. For ImageNet-1k, the diffusion model was trained from scratch for 12 epochs, followed by 8 epochs of ADDT fine-tuning, with guidance from a pretrained ResNet-101 classifier.

As shown in Table 8 and Table 9, ADDT improves the robustness of DBP on these complex datasets. However, the additional robustness provided by ADDT appears relatively modest. This suggests that, to scale ADDT more effectively, adequate model capacity and a sufficient volume of data are

Table 8: Clean and robust accuracy (%) on Tiny-ImageNet with WRN-28-10 classifier. ADDT improves DBP robustness on Tiny-ImageNet (-: classifier only).

| DBP method | Vanilla | | | ADDT | | |
|---|---|---|---|---|---|---|
| | $Clean$ | $\ell_\infty$ | $\ell_2$ | $Clean$ | $\ell_\infty$ | $\ell_2$ |
| - | 71.37 | 0.00 | 0.00 | - | - | - |
| DP$_{\text{DDPM}}$ | 57.13 | 11.82 | 46.68 | 56.15 | **13.57** | **48.54** |
| DP$_{\text{DDIM}}$ | 60.35 | 4.79 | 39.75 | 60.45 | **5.86** | **40.82** |
| DP$_{\text{EDM}}$ | 57.03 | 19.14 | 46.00 | 56.45 | **20.61** | **47.95** |

Table 9: Clean and robust accuracy (%) on ImageNet-1k with ResNet-101 classifier. Experiments are conducted under $\ell_\infty$ perturbation bound of $\epsilon = 4/255$.

| Metric | Vanilla | ADDT |
|---|---|---|
| Clean | 80.31 | 80.20 |
| PGD20+EoT10 | 46.92 | **48.02** |
| PGD200+EoT20 | 35.31 | **35.83** |

essential. This is further supported by the results in Table 3, where the larger DiffPure+ADDT model demonstrates greater robustness compared to the smaller DP$_{\text{DDPM}}$+ADDT model, aligning with findings from traditional AT (Wang et al., 2023; Huang et al., 2023).

It is also important to note that scaling to high-resolution images poses a significant challenge for attackers, as performing strong EoT attacks on high-resolution images is computationally expensive. For instance, our evaluation with PGD200+EoT20 on $224 \times 224$ images at a resolution of 1024 takes approximately 7 days of computation on 8 NVIDIA RTX 4090 GPUs.

### 6.5 DISCUSSIONS ON IMPROVING STOCHASTICITY-BASED DBP ROBUSTNESS

As discussed in Section 4.2, DBP robustness can be primarily attributed to the high variance of the stochastic attack gradients. We argue that *increasing the variance of attack gradients* can improve the stochasticity-based robustness of DBP models by reducing the effectiveness of EoT attacks. Specifically, on one hand, higher variance leads to larger errors in estimating the expected attack gradient direction. To reduce these errors, more EoT steps are required. On the other hand, higher variance also means that the DW-box attack gradient (which indicates the most effective attack direction) deviates more from the EoT attack gradient, even if the estimation of the mean attack gradient is accurate. As discussed in Section 4.2, this deviation leads to a lower increase in classification loss after one attack step, suggesting that a successful attack may not be achieved or require more PGD steps.

To increase the variance of attack gradients, a natural approach is to introduce more stochasticity. In an initial experiment, we augment the DBP framework's stochasticity by integrating a *Corrector sampler*. Specifically, Song et al. (2021b) propose a Predictor-Corrector (PC) sampler framework. While standard VPSDE (DDPM++) implementations typically use only the predictor component, we add a Corrector sampler to increase stochasticity in the reverse diffusion process, thereby enhancing the overall variance of attack gradients. As detailed in Appendix J, our preliminary results indicate that this modification improves the robustness of DBP models against adaptive White-box attacks. However, there is a trade-off: the model's clean accuracy decreases slightly. These observations align with the findings of Nie et al. (2022), where randomizing the diffusion timesteps also improves robustness at the expense of clean accuracy, as well as with previous research on stochastic preprocessing defenses (Gao et al., 2022).

## 7 CONCLUSION

This study provides a new perspective on the robustness of Diffusion-Based Purification (DBP), highlighting the critical role of stochasticity and challenging the traditional view that robustness primarily arises from minimizing the distribution gap through the forward diffusion process. We introduce a Deterministic white-box (DW-box) attack setting and demonstrate that DBP models rely on stochastic elements to evade effective attack directions but lack the ability to purify adversarial perturbations. This shows distinct differences compared to models trained with Adversarial Training. To further improve the robustness of DBP models, we propose Adversarial Denoising Diffusion Training (ADDT) and Rank-Based Gaussian Mapping (RBGM). ADDT integrates adversarial perturbations into the training process, while RBGM trims perturbations to better approximate Gaussian distributions. Experiments across various diffusion methods, attack settings, and datasets confirm the effectiveness of ADDT in enhancing robustness. In conclusion, this study emphasizes the decoupling of stochasticity-based and purification-based robustness of DBP models for deeper analysis, and suggests that combining both approaches can lead to improved robustness in practice.

## ACKNOWLEDGEMENTS

This work is supported in part by National Natural Science Foundation of China (NSFC) under Grant No.62376292, U21A20470, Guangdong Provincial General Fund No. 2024A1515010208.

We would like to express our sincere gratitude to **Hongjun Wang**, **Zhaoyu Chen**, **Huanran Chen**, **Ruoxin Chen** and **Conghan Yue** for their invaluable discussions, which greatly contributed to the development of this work.

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

# A    INFLUENCE OF EoT ITERATIONS ON DBP ROBUSTNESS EVALUATION

In this section, we examine how the number of EoT iterations influences the DBP robustness evaluation. As previously discussed in Section 4.1, the Deterministic White-box attack could find the most effective attack direction. To quantify the impact of EoT iterations, we compare the attack direction of the standard White-box-EoT across various numbers of EoT iterations with that of the Deterministic White-box.

See Figure 7 for a visual explanation, where the red line shows the classification accuracy, and the blue line shows the similarity between the attack directions of the White-box-EoT and Deterministic White-box. The results show a clear trend: increasing the EoT iterations raises the similarity between the attack directions and reduces model accuracy.

Note that both the increase in similarity and the decline in accuracy converge as the number of iterations increases. Balancing computational cost and evaluation accuracy, we adopted the PGD20-EoT10 configuration for our robustness evaluation.

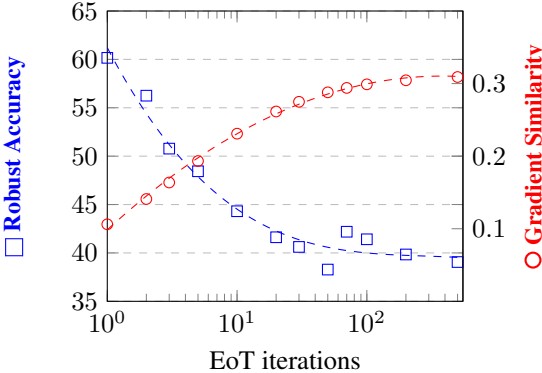

Figure 7: Robust accuracy (%) and gradient similarity on $DP_{DDPM}$ for CIFAR-10, obtained by different EoT iterations. As the number of EoT iterations increases, the gradient similarity between the White-box-EoT attack direction and the Deterministic White-box attack direction increases and the robust accuracy decreases.

# B    DBP MODELS WITH DIFFERENT STOCHASTIC ELEMENTS CANNOT BE ATTACKED SIMULTANEOUSLY

Previous research has raised concerns about whether stochasticity enhances robustness, suggesting it may create obfuscated gradients that provide a false sense of security (Athalye et al., 2018a). To investigate this, we implement $DW_{Semi}$-box, a semi-stochastic framework that restricts the available stochastic elements to a limited set of options, in order to explore whether stochasticity can genuinely improve robustness.

$DW_{semi-128}$ builds on the concept of Deterministic White-box. In contrast to the Deterministic White-box approach, where the attacker exploits the exact stochastic noise used during evaluation, $DW_{semi-128}$ limits the stochastic elements to a constrained set of possibilities. To mount an attack across 128 distinct sets of stochastic noise, the attacker can employ the average adversarial direction across these 128 noise settings (EoT-128) to perturb the DBP model. We evaluate the impact of stochasticity by comparing the loss changes under DW-box and $DW_{semi-128}$ attacks. This is done by adjusting a factor $k$ to modify an image $x$ with a perturbation $\sigma$, and evaluating the loss at $x + k\sigma$ for $k$ ranging from $-16$ to $16$. Perturbations are generated using the $\ell_\infty$ Fast Gradient Sign Method (FGSM) (Goodfellow et al., 2015) with a magnitude of $1/255$. The experiment is conducted using the WideResNet-28-10 with $DP_{DDPM}$ over the first 128 images from the CIFAR-10 dataset.

As shown in Figure 8, in the Deterministic White-box setting, perturbations lead to a significant increase in loss, demonstrating their effectiveness. However, under $DW_{semi-128}$, the loss increase is more moderate. This suggests that even when stochastic elements are limited to a small set of

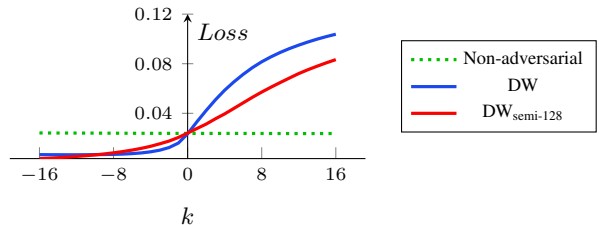

Figure 8: Impact of stochasticity on perturbation efficacy. Perturbations created under the $\text{DW}_\text{semi}$-box setting are less effective compared to those under the DW-box setting. For non-adversarial perturbations, each element is randomly assigned a value of either $1/255$ or $-1/255$.

possibilities, and the attacker has fully considered all options, stochasticity still contributes to the robustness of the DBP model. These results contradict the hypothesis that there might be a universally vulnerable direction when DBP applies different stochastic elements, thereby challenging concerns about potential insecurity.

## C  IMPACT OF ATTACKERS' KNOWLEDGE ON ROBUSTNESS: COMPARISON OF ATTACK SETTINGS

This section explores how the robustness of diffusion-based models is affected by the levels of knowledge that attackers possess concerning stochastic components in diffusion processes. We analyze how the knowledge of forward and reverse diffusion processes impacts model robustness in various attack settings.

### C.1  STOCHASTIC ELEMENTS IN THE DIFFUSION PROCESSES

Unlike deterministic models, where outputs are solely determined by the inputs, diffusion models introduce stochastic elements that also influence the outcomes. To explain the attacker's knowledge, we first explain the stochastic components involved in diffusion processes.

In the **forward diffusion process**, Gaussian noise is incorporated into the input data to derive a noisy version $\boldsymbol{x}_t$:

$$\boldsymbol{x}_t = \sqrt{\bar{\alpha}_t}\,\boldsymbol{x} + \sqrt{1 - \bar{\alpha}_t}\,\boldsymbol{\epsilon}_f, \tag{10}$$

where $\boldsymbol{\epsilon}_f \sim \mathcal{N}(\mathbf{0}, \boldsymbol{I})$ is sampled once per input.

In the **reverse diffusion process**, the model progressively denoises $\boldsymbol{x}_t$ through iterative steps. For the Denoising Diffusion Probabilistic Model (DDPM), the reverse process is inherently stochastic:

$$\boldsymbol{x}_{t-1} = \frac{1}{\sqrt{\alpha_t}}\left(\boldsymbol{x}_t - \frac{1 - \alpha_t}{\sqrt{1 - \bar{\alpha}t}}\boldsymbol{\epsilon_\theta}(\boldsymbol{x}_t, t)\right) + \sigma_t \boldsymbol{\epsilon}_t, \tag{11}$$

where $\boldsymbol{\epsilon}_t \sim \mathcal{N}(0, I)$ is sampled at each reverse step. In contrast, for the Denoising Diffusion Implicit Model (DDIM), the reverse process is deterministic, and no noise $\{\boldsymbol{\epsilon}_t\}_{t=1}^T$ is added.

### C.2  ATTACK SETTINGS AND ATTACKER KNOWLEDGE

We define four attack settings based on the information accessible to the attacker, particularly regarding the Gaussian noise variables involved in the diffusion process. Table 10 provides a summary of the attacker's knowledge in each setting.

In the conventional white-box attack setting, the attacker possesses comprehensive knowledge of the model architecture and parameters but lacks insight into the stochastic elements used during inference ($\boldsymbol{\epsilon}_f$ and $\{\boldsymbol{\epsilon}_t\}_{t=1}^T$). The $\text{DW}_\text{Fwd}$ setting grants the attacker knowledge of the Gaussian noise in the forward diffusion process ($\boldsymbol{\epsilon}_f$). Conversely, the $\text{DW}_\text{Rev}$ setting provides the attacker with knowledge of the Gaussian noise introduced during the reverse diffusion steps ($\{\boldsymbol{\epsilon}_t\}_{t=1}^T$). The $\text{DW}_\text{Both}$ setting offers the attacker complete access to all stochastic elements, $\boldsymbol{\epsilon}_f$ and $\{\boldsymbol{\epsilon}_t\}_{t=1}^T$. By manipulating the

Table 10: Information accessible to the attacker in different attack settings. $\epsilon_f$ denotes the Gaussian noise in the forward process, and $\{\epsilon_t\}_{t=1}^T$ represents the Gaussian noise in the reverse process.

| Attacker's Knowledge | White-box | $DW_{Fwd}$ | $DW_{Rev}$ | $DW_{Both}$ |
|---|---|---|---|---|
| Model Architecture and Parameters | ✓ | ✓ | ✓ | ✓ |
| Input Images and Class Labels | ✓ | ✓ | ✓ | ✓ |
| Forward Process Noise $\epsilon_f$ | ✗ | ✓ | ✗ | ✓ |
| Reverse Process Noise $\{\epsilon_t\}_{t=1}^T$ | ✗ | ✗ | ✓ | ✓ |

attacker's knowledge in this manner, we isolate the individual effects of the forward and reverse diffusion processes on model robustness.

### C.3  IMPLICATIONS OF THE ATTACKER'S KNOWLEDGE OF STOCHASTIC ELEMENTS

The attacker's capability to craft potent adversarial examples is significantly influenced by their knowledge of the stochastic elements in diffusion processes. When these elements are unknown to the attacker, they must independently sample noise variables, leading to discrepancies between their approximations and the actual behavior of the victim model. Conversely, if the attacker has access to the exact noise variables used during inference, they can accurately replicate the model's behavior, significantly enhancing the effectiveness of their attack.

**Attacker Without Knowledge of Stochastic Elements.**   In scenarios where the attacker lacks access to the specific noise variables $\epsilon_f$ and $\{\epsilon_t\}_{t=1}^T$, the model's output becomes unpredictable from the attacker's perspective. Consequently, the attacker must optimize the expected value of the loss function over the distribution of these stochastic elements. The optimization problem for generating an adversarial example $x^{adv}$ is formulated as:

$$x^{adv} = \arg\max_{\|x^{adv}-x\|\leq\delta} \mathbb{E}_{\epsilon_f,\{\epsilon_t\}_{t=1}^T} \left[ L\left(f(x^{adv};\epsilon_f,\{\epsilon_t\}_{t=1}^T),y\right)\right], \tag{12}$$

where $\delta$ specifies the permissible perturbation magnitude, $L$ is the loss function, $f$ represents the model's output given the input and stochastic elements, and $y$ is the actual class label.

**Attacker With Knowledge of Stochastic Elements.**   If the attacker possesses precise knowledge of the noise variables $\epsilon_f$ and $\{\epsilon_t\}_{t=1}^T$ used during the model's inference, they can accurately replicate the behavior of the victim classifier. In this case, the stochastic processes become deterministic from the attacker's perspective, enabling the optimization problem to be formulated as:

$$x^{adv} = \arg\max_{\|x^{adv}-x\|\leq\delta} L\left(f(x^{adv};\epsilon_f,\{\epsilon_t\}_{t=1}^T),y\right). \tag{13}$$

This precise knowledge allows the attacker to adopt the exact noise that will be used during the target evaluation, allowing effective evaluation.

### C.4  EFFECT OF ATTACKER'S KNOWLEDGE ON MODEL ROBUSTNESS

We test the robustness of DDPM under these four settings, and Table 11 encapsulates the result.

Table 11: Robust accuracy (%) of DDPM under different attack settings.

| Attack Setting | Robust Accuracy ($\ell_\infty$) |
|---|---|
| Conventional White-Box Attack | 47.27 |
| $DW_{Fwd}$ | 45.41 |
| $DW_{Rev}$ | 35.25 |
| $DW_{Both}$ | 16.80 |

**Conventional White-Box Attack.**   In this setting, the attacker has full knowledge of the model's architecture and parameters but lacks access to the stochastic components ($\epsilon_f$ and $\{\epsilon_t\}_{t=1}^T$) involved during inference. Hence, the model's output is unpredictable due to the stochasticity of both diffusion processes, making it challenging for the attacker to generate effective adversarial examples (reaching robust accuracy of **47.27%**).

**DW$_{\text{Fwd}}$.** Here, the attacker is aware of the Gaussian noise $\epsilon_f$ used in the forward diffusion process but does not have knowledge of the noise $\{\epsilon_t\}_{t=1}^T$ used in the reverse process. This partial information enables the attacker to replicate the forward process, reducing uncertainty. However, the reverse process remains unpredictable. The exposure of the forward process leads to a slight decrease in robust accuracy to **45.41%**.

**DW$_{\text{Rev}}$.** In this case, the attacker knows the noise variables $\{\epsilon_t\}_{t=1}^T$ used in the reverse diffusion process but lacks knowledge of the forward process noise $\epsilon_f$. This enables the attacker to align their strategy more closely with the behavior of the model during the reverse diffusion phase, resulting in a more substantial drop in robust accuracy to **35.25%**. These findings suggest that the stochasticity of the reverse process plays a more critical role in maintaining model robustness than that of the forward process.

**DW$_{\text{Both}}$.** In this case, the attacker possesses full knowledge of the noise variables for both the forward and reverse diffusion processes, allowing them to precisely replicate both processes and eliminate stochasticity from their perspective. This complete predictability enables the attacker to craft highly effective adversarial examples, leading to a significant reduction in robust accuracy to **16.80%**. These results underscore the importance of the stochastic elements in preserving model robustness; when fully exposed, the model's defense mechanisms are substantially weakened.

## D   THE ROLE OF STOCHASTICITY IN DBP COMPARED TO CERTIFIED DEFENSE METHODS

In this appendix section, we delve deeper into the role of stochasticity in Diffusion-Based Prediction (DBP) models and contrast it with its role in certified defense methods such as randomized smoothing (Cohen et al., 2019). While both approaches incorporate stochasticity, their mechanisms and implications for adversarial robustness differ significantly.

- Conventionally, the classification models discussed in the studies of adversarial robustness can be viewed as mappings from input space $\mathcal{X}$ to the label space $\mathcal{Y}$. However, DBP additionally involves a random variable $\epsilon \in \mathcal{E}$ that determines the random sampling in the forward and reverse processes (which can be the random seed in implementation). Hence, a DBP model protected classifier $f$ can be viewed as the mapping $f : (\mathcal{X}, \mathcal{E}) \to \mathcal{Y}$.

- Previous studies on randomized smoothing treat the randomized model $f$ as a mapping $f : \mathcal{X} \to \mathcal{P}_{\mathcal{Y}}$, where $\mathcal{P}_{\mathcal{Y}}$ is the space of label distribution. Typically, the final prediction can be formulated as $F(\boldsymbol{x}) = \arg\max_c [f(\boldsymbol{x})]_c$, i.e., the class $c$ with the highest probability in the output distribution $\mho(\boldsymbol{x})$. Apparently, $F$ deterministically maps $\mathcal{X}$ to $\mathcal{Y}$, consistent with the conventional models.

- Recent studies on DBP also regard the model as $f : \mathcal{X} \to \mathcal{P}_{\mathcal{Y}}$, without explicitly studying the role of $\epsilon$. *The key difference between DBP and randomized smoothing is that the final prediction for an input $\boldsymbol{x}$ is directly sampled from the distribution $f(\boldsymbol{x})$ for once, instead of sampling multiple times to approximate $F(\boldsymbol{x})$ as in randomized smoothing.*

- In this paper, we revisit DBP by treating the randomized model $f$ as the mapping $f : \mathcal{X}, \mathcal{E}) \to \mathcal{Y}$ and studying the role of $\epsilon \in \mathcal{E}$ as an input of $f$. From this perspective, the conventional adversarial setting assuming full knowledge of the model parameters (but not $\epsilon$) is not a complete white box, which motivates us to study the DW-box setting.

- From our perspective, we can clearly point out the difference between DBP and randomized smoothing in terms of the loss landscape. Given an input $\boldsymbol{x}_0$, the local loss landscape for a DBP model $f$ is not deterministic as it also depends on $\epsilon$. *Although the expected loss landscape over $\epsilon \in \mathcal{E}$ may be smooth, it does not suggest the robustness of DBP, as $\epsilon$ is fixed during a single inference run of DBP.* Indeed, our study suggests that given $\boldsymbol{x}_0$ and a fixed $\epsilon_0$, the local landscape of DBP is likely not smooth. In contrast, the loss landscape of a randomized smoothing model $F$ may be smooth as it is the average landscape over multiple $\epsilon$. To conclude, we argue that the random noise itself may not smooth the loss landscape, but the average over random noises may.

# E    EVALUATION SETTINGS

Previous assessments of DBP robustness have often utilized potentially unreliable methods. In particular, due to the iterative denoising process in diffusion models, some studies resort to mathematical approximations of gradients to reduce memory constraints (Nie et al., 2022) or to circumvent the diffusion process during backpropagation (Wang et al., 2022). Furthermore, the reliability of AutoAttack, a widely used evaluation method, in assessing the robustness of DBP models is questionable. Although AutoAttack includes a *Rand* version designed for stochastic models, Nie et al. (2022) have found instances where the *Rand* version is less effective than the *Standard* version in evaluating DBP robustness.

To improve the robustness evaluation of diffusion-based purification (DBP) models, we implement several modifications. First, to ensure the accuracy of the gradient computations, we compute the exact gradient of the entire diffusion classification pipeline. To mitigate the high memory requirements in diffusion iterative denoising steps, we use gradient checkpointing (Chen et al., 2016) techniques to optimize memory usage. In addition, to deal with the stochastic nature of the DBP process, we incorporate the Expectation over Transformation (EoT) method to average gradients across different attacks. We adopt EoT with 10 iterations, and a detailed discussion of the choice of EoT iterations can be found in Appendix A. We also use the Projected Gradient Descent (PGD) attack instead of AutoAttack for our evaluations. We discover a bug in the *Rand* version of AutoAttack that causes it to overestimate the robustness of DBP. After fixing this, AutoAttack$_{Fixed}$ gives similar results to PGD attacks, but at a much higher computational cost. Our revised robustness evaluation revealed that DBP models, such as DiffPure and GDMP, perform worse than originally claimed. DiffPure's accuracy dropped from a claimed 70.64% to an actual 55.96%, and GDMP's from 90.10% to 40.97%. These results emphasize the urgent need for more accurate and reliable evaluation methods to properly assess the robustness of DBP models. Similar evaluation protocols are also applied in Chen et al. (2023); Kang et al. (2024).

## E.1    EVALUATION WITH FIXED AUTOATTACK

AutoAttack (Croce & Hein, 2020), an ensemble of White-box and Black-box attacks, is a popular benchmark for evaluating model robustness. It is used in RobustBench (Croce et al., 2020) to evaluate over 120 models. However, Nie et al. (2022) finds that the *Rand* version of AutoAttack, designed to evaluate stochastic defenses, sometimes yields higher accuracy than the *Standard* version that is intended for deterministic methods. Our comparison of AutoAttack and PGD20-EoT10 in Table 12 also shows that the *Rand* version of AutoAttack gives higher accuracy than the PGD20-EoT10 attack.

We attribute this to the sample selection of AutoAttack. As an ensemble of attack methods, AutoAttack selects the final adversarial sample from either the original input or the attack results. However, the original implementation neglects stochasticity and considers an adversarial sample to be sufficiently adversarial if it gives a false result in one evaluation. To fix this, we propose a 20-iteration evaluation and select the adversarial example with the lowest accuracy. The flawed code is in the official GitHub main branch, git version $a39220048b3c9f2cca9a4d3a54604793c68eca7e$, and specifically in lines #125, #129, #133-136, #157, #221-225, #227-228, #231 of the file 'autoattack/autoattack.py'. We will open-source our updated code and encourage future stochastic defense methods to be evaluated against the fixed code. The code can now be found at: https://github.com/Buntender/auto-attack.

Following the fix, robust accuracy under AutoAttack$_{Fixed}$ decreases by up to 10 points, producing results comparable to our PGD20-EoT10 test outcomes. However, using AutoAttack on DP$_{DDPM}$ took nearly 25 hours, five times longer than PGD20-EoT10. Therefore, we will use PGD20-EoT10 for most robustness evaluation.

# F    EXPERIMENTAL SETTING OF VISUALIZATION OF THE ATTACK TRAJECTORY

We visualize the attack by plotting the loss landscape and tracing the trajectories of EoT attack under White-box setting and the Deterministic White-box setting in Figure 3. We run a vanilla PGD20-EoT10 attack under White-box setting and a PGD20 attack under Deterministic White-box setting. We then expand a 2D space using the final perturbations from these two attacks, draw the loss landscape, and plot the attack trajectories on it. Note that the two adversarial perturbation directions

Table 12: AutoAttack (*Rand* version) and PGD20-EoT10 performance on DBP methods for CIFAR-10 (the lower the better). The original AutoAttack produces high accuracy (%), after fixing, it achieves similar results to PGD20+EoT10 attack.

| DBP method | $\ell_\infty$ | | | $\ell_2$ | | |
|---|---|---|---|---|---|---|
| | AutoAttack | AutoAttack$_{\text{Fixed}}$ | PGD20-EoT10 | AutoAttack | AutoAttack$_{\text{Fixed}}$ | PGD20-EoT10 |
| DiffPure | 62.11 | 56.25 | **55.96** | 81.84 | 76.37 | **75.78** |
| DP$_{\text{DDPM}}$ | 57.81 | **46.88** | 48.63 | 71.68 | **71.09** | 72.27 |
| DP$_{\text{DDIM}}$ | 50.20 | **40.62** | 44.73 | 77.15 | **70.70** | 71.68 |

Table 13: Clean and robust accuracy (%) on different DBP methods for CIFAR-10, evaluated with AutoAttack$_{\text{Fixed}}$ (*Rand* version). All methods show consistent improvement when fine-tuned with ADDT.

| DBP method | Vanilla | | | ADDT | | |
|---|---|---|---|---|---|---|
| | *Clean* | $\ell_\infty$ | $\ell_2$ | *Clean* | $\ell_\infty$ | $\ell_2$ |
| DiffPure | 89.26 | 56.25 | 76.37 | 89.94 | **58.20** | **77.34** |
| DP$_{\text{DDPM}}$ | 85.94 | 46.88 | 71.09 | 85.64 | **48.63** | **72.27** |
| DP$_{\text{DDIM}}$ | 88.38 | 40.62 | 70.70 | 88.77 | **44.73** | **71.68** |

are not strictly orthogonal. To extend this 2D space, we use the Deterministic White-box attack direction and the orthogonal component of the EoT attack direction. Note that the endpoints of both trajectories lie exactly on the loss landscape, while intermediate points are projected onto it. The plot is evaluated using WideResNet-28-10 with DP$_{\text{DDPM}}$ over the first 128 images of CIFAR-10 dataset.

## G    PSEUDO-CODE OF ADDT

The pseudo-code for adopting ADDT within DDPM and DDIM framework is shown in Algorithm 1.

---

**Algorithm 1** Adversarial Denoising Diffusion Training (ADDT)

---

**Require:** $\boldsymbol{x}_0$ is image from training dataset, $y$ is the class label of the image, $C$ is the classifier, $\boldsymbol{P}$ is one-step diffusion reverse process and $\boldsymbol{\theta}$ is it's parameter, $L$ is CrossEntropy Loss.
1: **for** $\boldsymbol{x}_0$, $y$ in the training dataset **do**
2:      $t \sim \mathcal{U}(\{1, \ldots, T\})$
3:      $\lambda_t = \texttt{clip}(\gamma_t \lambda_{\text{unit}}, \lambda_{\min}, \lambda_{\max})$, where $\gamma_t = \frac{\sqrt{\overline{\alpha}_t}}{\sqrt{1-\overline{\alpha}_t}}$
4:      Init $\boldsymbol{\delta}$ to a small random vector.
5:      **for** 1 to ADDT$_{\text{iterations}}$ **do**
6:          $\boldsymbol{\epsilon} \sim \mathcal{N}(0, I)$
7:          $\boldsymbol{\epsilon}' = \texttt{RBGM}(\boldsymbol{\delta}, \boldsymbol{\epsilon})$
8:          $\boldsymbol{x}_t = \sqrt{\overline{\alpha}_t}\boldsymbol{x}_0 + \sqrt{1-\lambda_t^2}\sqrt{1-\overline{\alpha}_t}\boldsymbol{\epsilon} + \lambda_t\sqrt{1-\overline{\alpha}_t}\boldsymbol{\epsilon}'$
9:          $\boldsymbol{\delta} = \boldsymbol{\delta} + \nabla_{\boldsymbol{\epsilon}'} L(C(\boldsymbol{P}(\boldsymbol{\theta}, \boldsymbol{x}_t, t), y))$
10:     **end for**
11:      $\boldsymbol{\epsilon} \sim \mathcal{N}(0, I)$
12:      $\boldsymbol{\epsilon}' = \texttt{RBGM}(\boldsymbol{\delta}, \boldsymbol{\epsilon})$
13:      $\boldsymbol{x}_t = \sqrt{\overline{\alpha}_t}\boldsymbol{x}_0 + \sqrt{1-\lambda_t^2}\sqrt{1-\overline{\alpha}_t}\boldsymbol{\epsilon} + \lambda_t\sqrt{1-\overline{\alpha}_t}\boldsymbol{\epsilon}'$
14:      Take a gradient descent step on:
         $\nabla_{\boldsymbol{\theta}} \| \frac{\sqrt{\overline{\alpha}_t}}{\sqrt{1-\overline{\alpha}_t}}(\boldsymbol{x}_0 - \boldsymbol{P}(\boldsymbol{\theta}, \boldsymbol{x}_t, t)) \|_2^2$
15: **end for**
     Diffusion model $\boldsymbol{\epsilon_\theta}$ predicts the Gaussian noise added to the image, adopting Equation (4) in the paper, we have $\boldsymbol{P}(\boldsymbol{\theta}, \boldsymbol{x}_t, t) = \left(\boldsymbol{x}_t - \sqrt{1-\overline{\alpha}_t}\boldsymbol{\epsilon_\theta}(\boldsymbol{x}_t, t)\right) / \sqrt{\overline{\alpha}_t}$

---

## H    ADDT RESULTS ON DP$_{\text{DDIM}}$

As shown in Table 14, the performance of DP$_{\text{DDIM}}$ is less sensitive to the number of function evaluations (NFEs). Additionally, ADDT consistently improved the robustness of DP$_{\text{DDIM}}$.

Table 14: Clean and robust accuracy (%) on DP$_{\text{DDIM}}$. ADDT improve robustness across different NFEs (*: default DDIM generation setting, -: classifier only ).

| Dataset | NFEs | Vanilla | | | ADDT | | |
|---|---|---|---|---|---|---|---|
| | | $Clean$ | $\ell_\infty$ | $\ell_2$ | $Clean$ | $\ell_\infty$ | $\ell_2$ |
| CIFAR-10 | - | 95.12 | 0.00 | 1.46 | 95.12 | 0.00 | 1.46 |
| | 5 | 89.65 | 42.19 | 68.65 | 88.57 | **47.27** | **70.61** |
| | 10* | 88.96 | 43.16 | 67.58 | 88.18 | **47.85** | **70.61** |
| | 20 | 87.89 | 41.70 | 69.24 | 88.67 | **48.63** | **69.73** |
| | 50 | 88.96 | 42.48 | 68.85 | 88.57 | **46.68** | **69.24** |
| | 100 | 88.38 | 42.19 | 70.02 | 88.77 | **46.48** | **71.19** |
| CIFAR-100 | - | 76.66 | 0.00 | 2.44 | 76.66 | 0.00 | 2.44 |
| | 5 | 62.11 | 15.43 | 35.74 | 62.79 | **17.58** | **38.87** |
| | 10* | 62.21 | 15.33 | 36.52 | 64.45 | **20.02** | **39.26** |
| | 20 | 63.67 | 15.62 | 37.89 | 65.23 | **18.65** | **40.62** |
| | 50 | 62.40 | 16.31 | 37.79 | 63.87 | **19.14** | **39.94** |
| | 100 | 63.28 | 15.23 | 36.62 | 66.02 | **18.85** | **39.84** |

# I  ADOPTING VPSDE(DDPM++) AND EDM MODELS IN DBP

In the previous discussion of the robustness of DBP models, as detailed in Section 4.1, our focus was primarily on the DDPM and DDIM models. We now extend our analysis to include VPSDE (DDPM++) and EDM (Karras et al., 2022) models. VPSDE (DDPM++) is the diffusion model used in DiffPure.

From a unified perspective, the diffusion process can be modeled by stochastic differential equations (SDE) (Song et al., 2021b). The forward SDE, as described in Equation (14), converts a complex initial data distribution into a simpler, predetermined prior distribution by progressively infusing noise. This can also be done in a single step, as shown in Equation (15), mirroring the strategy of DDPM described in Equation (2). Reverse SDE, as explained in Equation (16), reverses this process, restoring the noise distribution to the original data distribution, thus completing the diffusion cycle.

$$\mathrm{d}\boldsymbol{x} = \boldsymbol{f}(\boldsymbol{x}, t)\mathrm{d}t + g(t)\mathrm{d}\boldsymbol{w}, \tag{14}$$

$$p_{0t}(\boldsymbol{x}(t) \mid \boldsymbol{x}(0)) =$$
$$\mathcal{N}\left(\boldsymbol{x}(t); e^{-\frac{1}{4}t^2(\bar{\beta}_{\max} - \bar{\beta}_{\min}) - \frac{1}{2}t\bar{\beta}_{\min}}\boldsymbol{x}(0), \boldsymbol{I} - \boldsymbol{I}e^{-\frac{1}{2}t^2(\bar{\beta}_{\max} - \bar{\beta}_{\min}) - t\bar{\beta}_{\min}}\right), \quad t \in [0, 1] \tag{15}$$

$$\mathrm{d}\boldsymbol{x} = \left[\boldsymbol{f}(\boldsymbol{x}, t) - g^2(t)\nabla_{\boldsymbol{x}} \log p_t(\boldsymbol{x})\right]\mathrm{d}t + g(t)\mathrm{d}\bar{\boldsymbol{w}}. \tag{16}$$

The reverse process of SDEs also derives equivalent ODEs Equation (17) for fast sampling and exact likelihood computation, and these Score ODEs correspond to DDIM.

$$\mathrm{d}\boldsymbol{x} = \left[\boldsymbol{f}(\boldsymbol{x}, t) - \frac{1}{2}g(t)^2\nabla_{\boldsymbol{x}} \log p_t(\boldsymbol{x})\right]\mathrm{d}t. \tag{17}$$

By modulating the stochasticity, we can craft a spectrum of semi-stochastic models that bridge pure SDEs and deterministic ODEs, offering a range of stochastic behaviors.

EDM provides a unified framework to synthesize the design principles of different diffusion models (DDPM, DDIM,iDDPM (Nichol & Dhariwal, 2021), VPSDE, VESDE (Song et al., 2021b)). Within this framework, EDM incorporates efficient sampling methods, such as the Heun sampler, and introduces optimized scheduling functions $\sigma(t)$ and $s(t)$. This allows EDM to achieve state-of-the-art performance in generative tasks.

EDM forward process could be presented as:

$$\boldsymbol{x}_t = \boldsymbol{x}_0 + \sigma(t^*)\boldsymbol{\epsilon}, \quad \boldsymbol{\epsilon} \sim \mathcal{N}(\boldsymbol{0}, \boldsymbol{I}), \tag{18}$$

where we choose $\sigma(t^*) = 0.5$ for clean and robust accuracy tradeoff. And for the reverse process, EDM incorporates a parameter $S_{churn}$ to modulate the stochastic noise infused during the reverse

process. For our experiments, we choose 50 reverse steps (50 NFEs, NFEs is Function of Neural Function Evaluations), configured the parameters with $S_{min} = 0.01$, $S_{max} = 0.46$, $S_{noise} = 1.007$, and designate $S_{churn} = 0$ to represent EDM-ODE, $S_{churn} = 6$ to represent EDM-SDE.

As shown in Table 15, our ADDT could also increase the robustness of $DP_{EDM}$.

Table 15: Clean and robust accuracy (%) on $DP_{EDM}$ for CIFAR-10. ADDT improves robustness in both $DP_{EDM\text{-}SDE}$ and $DP_{EDM\text{-}ODE}$.

| Metric | Vanilla | | ADDT | |
|---|---|---|---|---|
| | $DP_{EDM\text{-}SDE}$ | $DP_{EDM\text{-}ODE}$ | $DP_{EDM\text{-}SDE}$ | $DP_{EDM\text{-}ODE}$ |
| $Clean$ | 86.43 | 87.99 | 86.33 | 87.99 |
| $\ell_\infty$ | 62.50 | 60.45 | **66.41** | **64.16** |
| $\ell_2$ | 76.86 | 75.49 | **79.16** | **77.15** |

## J    STRENGTHENING DBP VIA AUGMENTED STOCHASTICITY

(Song et al., 2021b) present a Predictor-Corrector sampler for SDEs reverse process for VPSDE (DDPM++), as detailed in Appendix I. However, standard implementations of VPSDE (DDPM++) typically use only the Predictor. Given our hypothesis that stochasticity contributes to robustness, we expect that integrating the Corrector sampler into VPSDE (DDPM++) would further enhance the robustness of DBP models. Our empirical results, as shown in Table 16, confirm that the inclusion of a Corrector to VPSDE (DDPM++) indeed improves the model's defense ability against adversarial attacks with $\ell_\infty$ norm constraints. This finding supports our claim that the increased stochasticity can further strengthen DBP robustness. Adding Corrector is also consistent with ADDT. Note that the robustness against $\ell_2$ norm attacks does not show a significant improvement with the integration of the Extra Corrector. A plausible explanation for this could be that the robustness under $\ell_2$ attacks is already quite strong, and the compromised performance on clean data counteracts the increase in robustness.

Table 16: Clean and robust accuracy (%) on $DP_{DDPM++}$ for CIFAR-10. Both extra Corrector and ADDT fine-tuning improved robustness.

| Metric | Vanilla | Extra Corrector | ADDT | ADDT+Extra Corrector |
|---|---|---|---|---|
| $Clean$ | 89.26 | 85.25 | 89.94 | 85.55 |
| $\ell_\infty$ | 55.96 | 59.77 | 62.11 | **65.23** |
| $\ell_2$ | 75.78 | 74.22 | **76.66** | **76.66** |

## K    DISCUSSION ABOUT RBGM-MAPPED PERTURBATIONS

### K.1    MOTIVATION AND ADVANTAGES OF RBGM

To elaborate, a conventional diffusion forward process is based on the equation:

$$\boldsymbol{x}_t = \sqrt{\overline{\alpha}_t}\,\boldsymbol{x}_0 + \sqrt{1 - \overline{\alpha}_t}\,\boldsymbol{\epsilon}, \tag{19}$$

where $\boldsymbol{x}_t$ represents the noisy image at time $t$, $\boldsymbol{x}_0$ is the initial input, $\overline{\alpha}_t$ is a time-dependent scaling factor, and $\boldsymbol{\epsilon}$ is random Gaussian noise. Our proposed method, ADDT, modifies this equation to include an adversarial component:

$$\boldsymbol{x}_t = \sqrt{\overline{\alpha}_t}\,\boldsymbol{x}_0 + \sqrt{1 - \lambda_t^2}\,\sqrt{1 - \overline{\alpha}_t}\,\boldsymbol{\epsilon} + \lambda_t\,\sqrt{1 - \overline{\alpha}_t}\,\boldsymbol{\epsilon_\delta}(\boldsymbol{\delta}). \tag{20}$$

In this revised formulation, $\boldsymbol{\epsilon_\delta}(\boldsymbol{\delta})$ represents the adversarial perturbation, and $\lambda_t$ is a parameter that controls the blend between traditional and adversarial noise. The core objective of ADDT training is

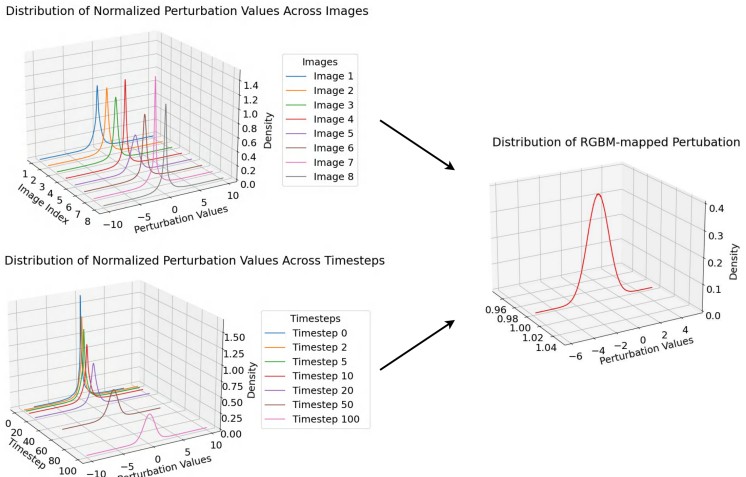

Figure 9: Raw perturbation values exhibit diverse distributions across images and time steps. RBGM maps these perturbations to a uniform Gaussian distribution.

to *generate perturbations that emulate the characteristics of Gaussian noise in conventional diffusion training while incorporating adversarial disturbances*. This leads to our Rank-Based Gaussian Mapping (RBGM) technique, which preserves the relative ordering of perturbation magnitudes while adjusting their values to more closely resemble a Gaussian distribution. The advantages of RBGM are twofold:

**Enhancing statistical consistency.** Raw adversarial perturbation values often exhibit non-standard distributions, and RBGM serves to recalibrate these perturbations, aligning them more closely with a Gaussian distribution. To elaborate, rather than enforcing a multivariate Gaussian distribution for the entire perturbation, RBGM ensures that the distribution of individual perturbation values adheres to Gaussian characteristics.

The benefit of this transformation can be illustrated in Figure 9 and Figure 10. For a fair comparison, the perturbation values have been normalized. In Figure 9, the original perturbation values display a wide array of distributions across different images and time steps. After the mapping of RBGM, these values are transformed to exhibit a uniform Gaussian distribution.

In Figure 10, the raw perturbations show irregular and inconsistent behavior when mixed with Gaussian noise at varying ratios. However, after RBGM mapping, the perturbations and the mixture exhibit consistent statistics with the pure Gaussian noise. The statistical consistency of the perturbation values may ease the training of the diffusion model and avoid significant deviation from the normal diffusion process.

**Reducing image-specific dependence.** In the training of diffusion models, the Gaussian noise is independent of specific images or time steps. This characteristic contrasts with adversarial perturbations, which are typically tailored to each input. RBGM addresses this issue by introducing stochasticity into the construction of perturbations while preserving the ranks of the values of the image-dependent adversarial perturbations. This approach effectively reduces image-specific dependence. Consequently, RBGM enhances ADDT's ability to better mimic the diffusion training process and potentially mitigates the overfitting of training images.

## K.2    RBGM-MAPPED PERTURBATIONS PRESERVE ADVERSARIAL CHARACTERISTICS

While RBGM-mapped perturbations are "selected from a Gaussian distribution", their actual distribution deviates from a pure Gaussian distribution and is adversarial for models. To substantiate

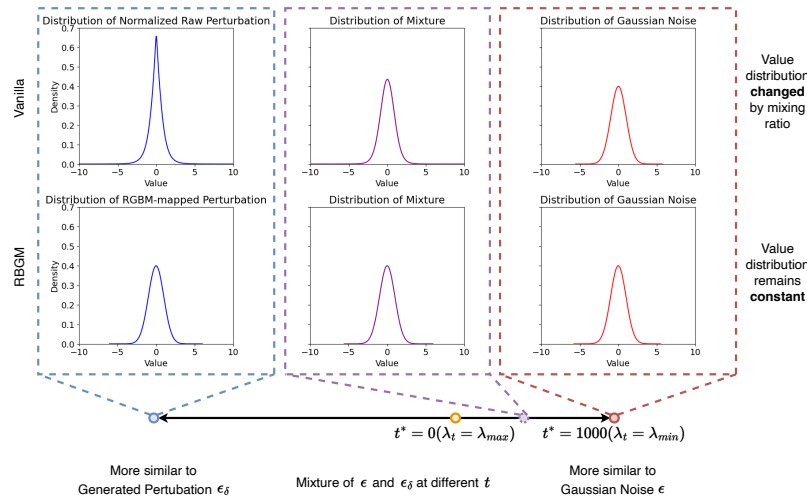

Figure 10: RBGM ensures that mixing perturbations with Gaussian noise at any ratio yields a consistent value distribution.

this claim, we compare the influence of RBGM-mapped perturbations and Gaussian noise on model performance. In our experiments, we perturb clean images by adding RBGM-mapped perturbations and Gaussian noise, each scaled by a factor of $0.03$. The results present in Table 17 demonstrate that RBGM-mapped perturbations effectively act as adversarial inputs to the model. These perturbations drastically reduce the accuracy of a pre-trained clean WRN-28-10 from $95.12\%$ to $4.47\%$.

Table 17: Comparison of model accuracy % under different conditions. RBGM-mapped perturbations lead to a significant reduction in accuracy compared to Gaussian noise.

| Classifier | Clean | Gaussian noise | RBGM-mapped perturbation |
|---|---|---|---|
| WRN-28-10 | 95.12 | 81.54 | 4.47 |

### K.3 Blending Adversarial Perturbations into Diffusion Model Training

In conventional adversarial attacks, perturbations are directly applied to the image, resulting in an adversarial image:

$$\boldsymbol{x}_{\text{adv}} = \boldsymbol{x}_0 + \boldsymbol{\delta}, \tag{21}$$

where $\boldsymbol{x}_0$ is the original input image, and $\boldsymbol{\delta}$ is the adversarial perturbation. In ADDT, we incorporate this concept into the diffusion process, redefining the noisy image at time step $t$ as:

$$\boldsymbol{x}_t = \sqrt{\overline{\alpha}_t}(\boldsymbol{x}_0 + \boldsymbol{\delta}) + \sqrt{1 - \overline{\alpha}_t}\boldsymbol{\epsilon}, \tag{22}$$

To enable the diffusion model to effectively purify adversarial perturbations during training, we reformulate the above equation by merging the perturbation $\boldsymbol{\delta}$ with the noise $\boldsymbol{\epsilon}$. This results in:

$$\boldsymbol{x}_t = \sqrt{\overline{\alpha}_t}\,\boldsymbol{x}_0 + \sqrt{1 - \overline{\alpha}_t}\,(\boldsymbol{\epsilon} + \gamma_t\,\boldsymbol{\delta}), \tag{23}$$

where $\gamma_t$ is a scaling factor defined as:

$$\gamma_t = \frac{\sqrt{\overline{\alpha}_t}}{\sqrt{1 - \overline{\alpha}_t}}. \tag{24}$$

Since $\overline{\alpha}_t$ is a time-dependent parameter that monotonically decreases from 1 to 0 during the diffusion process, $\gamma_t$ spans the range from 0 to $\infty$. To ensure the adversarial perturbation remains within a manageable intensity, we constrain its value to the range between $\lambda_{\min}$ and $\lambda_{\max}$.

## K.4 RBGM ENHANCES PERTURBATION COMPATIBILITY WITH DIFFUSION MODEL TRAINING

To illustrate how RBGM enhances the compatibility of perturbations with diffusion model training, we conduct comparative analyses in two scenarios. First, we assess the impact of RBGM on statistical consistency by comparing Gaussian noise with adversarial perturbations reordered based on Gaussian noise ranks. Second, we evaluate the effectiveness of RBGM-mapped perturbations in improving model robustness while maintaining generative performance by comparing them with $\ell_2$-normalized perturbations.

**Gaussian noise vs. adversarial perturbations ordered by Gaussian noise**    We begin by examining RBGM's influence on statistical consistency through two training methodologies:

1. **Vanilla**: Trained with standard Gaussian noise.

2. **ADDT$_{\text{Gaussian reordered}}$**: Trained with adversarial perturbations reordered according to Gaussian noise ranks. To ensure a fair comparison, the perturbations are normalized to have a mean of 0 and a variance of 1, as their original magnitudes (derived from accumulated gradients) are significantly smaller than those of standard Gaussian noise. Note that this approach—reordering adversarial perturbations based on Gaussian noise ranks—is distinct from RBGM, where Gaussian noise is reordered based on adversarial perturbation ranks.

The results presented in Table 18 reveal that models trained with Gaussian noise reordering using adversarial perturbation values exhibit lower accuracy on both clean and adversarial samples compared to vanilla models. This decline in performance underscores RBGM's ability to enhance perturbation compatibility with diffusion model training by improving statistical consistency.

Table 18: Comparison of DP$_{\text{DDPM}}$ accuracy under different conditions and perturbation types. Training with perturbations reordered by Gaussian noise and adversarial perturbation values degrades performance.

| NFEs | Vanilla | | | ADDT$_{\text{Gaussian reordered}}$ | | |
|---|---|---|---|---|---|---|
| | $Clean$ | $\ell_\infty$ | $\ell_2$ | $Clean$ | $\ell_\infty$ | $\ell_2$ |
| 5 | 49.51 | **21.78** | **36.13** | 48.40 | 21.10 | 33.40 |
| 10 | 73.34 | **36.72** | **55.47** | 71.78 | 34.07 | 52.98 |
| 20 | 81.45 | **45.21** | **65.23** | 79.99 | 42.43 | 64.21 |
| 50 | 85.54 | **46.78** | **68.85** | 83.90 | 46.73 | 68.17 |
| 100* | 85.94 | **47.27** | **69.34** | 84.54 | 46.98 | 69.33 |

**RBGM-mapped perturbations vs. $\ell_2$-normalized perturbations**    To further investigate RBGM's effectiveness in making adversarial perturbations compatible with diffusion model training, we compare:

1. **ADDT**: Trained with RBGM-mapped perturbations.

2. **ADDT$_{\ell_2\text{-normalized}}$**: Trained with raw adversarial perturbations, scaled to match the $\ell_2$ norm of standard Gaussian noise. This scaling ensures that the perturbations share the same $\ell_2$ norm as those mapped by RBGM, which we refer to as $\ell_2$-normalized perturbations.

As shown in Table 19, models trained with $\ell_2$-normalized perturbations tend to perform better under $\ell_2$ attacks in some scenarios (possibly because these perturbations are more similar to those generated by $\ell_2$ attack during testing). ADDT generally achieves better results. This advantage is particularly pronounced under $\ell_\infty$ attacks and in scenarios with higher NFEs. Furthermore, as shown in Table 20, ADDT yields a lower FID value, reflecting better preservation of generative capabilities.

As discussed in Appendix K.1, the goal of utilizing RGBM to generate perturbations is to mimic the characteristic of Gaussian noise, hence aligning ADDT closer to traditional diffusion model training. In this context, both RBGM and $\ell_2$ normalization serve as approximations of Gaussian noise. Yet, RBGM provides a more precise approximation, enhancing robustness and maintaining the generative performance more effectively than $\ell_2$ normalization.

Table 19: Comparison of $DP_{DDPM}$ accuracy under different perturbation conditions. Training with ADDT generally achieves better results. This advantage is particularly pronounced under $\ell_\infty$ attacks and in scenarios with higher NFEs.

| NFEs | ADDT | | | $ADDT_{\ell_2\text{-normalized}}$ | | |
|---|---|---|---|---|---|---|
| | Clean | $\ell_\infty$ | $\ell_2$ | Clean | $\ell_\infty$ | $\ell_2$ |
| 5 | 59.96 | **30.27** | 41.99 | 60.40 | 28.47 | **44.58** |
| 10 | 78.91 | **43.07** | 62.97 | 79.29 | 41.90 | **63.72** |
| 20 | 83.89 | **48.44** | **69.82** | 83.59 | 47.85 | 67.68 |
| 50 | 85.45 | **50.20** | 69.04 | 84.83 | 49.12 | **69.24** |
| 100* | 85.64 | **51.46** | **70.12** | 84.97 | 49.95 | 69.29 |

Table 20: FID score of DDPM for CIFAR-10 fine-tuned under different conditions. Fine-tuning with ADDT results in a lower FID score compared to $\ell_2$ normalization.

| | Clean fine-tuning | ADDT | $ADDT_{\ell_2\text{-normalized}}$ |
|---|---|---|---|
| FID | 3.50 | 5.190 | 5.678 |

## L    COMPARING RBGM-MAPPED PERTURBATIONS WITH $\ell_\infty$ PERTURBATIONS

In Section 6.3, we briefly explore the generation capabilities of diffusion models trained with RBGM-mapped and $\ell_\infty$ perturbations. Here, we provide further experiments and delve deeper into their robustness comparison. To train with $\ell_\infty$ perturbations, we adjust ADDT, replacing RBGM-mapped perturbations with $\ell_\infty$ perturbations. Here, instead of converting accumulated gradients to Gaussian-like perturbations, we use a 5-step projected gradient descent (PGD-5) approach. For fair comparison, we also set $\lambda_{unit} = 1, \lambda_{min} = 0, \lambda_{max} = 10$ and refer to this modified training protocol as $ADDT_{\ell_\infty}$.

We evaluate the clean and robust accuracy of ADDT and $ADDT_{\ell_\infty}$ fine-tuned models. These models exhibit different behaviors. As shown in Table 21, while Gaussian-mapped perturbations can simultaneously improve clean accuracy and robustness against both $\ell_2$ and $\ell_\infty$ attacks, training with $\ell_\infty$ perturbations primarily improves performance against $\ell_\infty$ attacks.

Table 21: Clean and robust accuracy (%) on DBP models trained with different perturbations for CIFAR-10. While ADDT simultaneously improves clean accuracy and robustness against both $\ell_2$ and $\ell_\infty$ attacks. $ADDT_{\ell_\infty}$ primarily improves performance against $\ell_\infty$ attacks.

| DBP method | Dataset | Vanilla | | | ADDT | | | $ADDT_{\ell_\infty}$ | | |
|---|---|---|---|---|---|---|---|---|---|---|
| | | Clean | $\ell_\infty$ | $\ell_2$ | Clean | $\ell_\infty$ | $\ell_2$ | Clean | $\ell_\infty$ | $\ell_2$ |
| $DP_{DDPM}$ | CIFAR-10 | 85.94 | 47.27 | 69.34 | 85.64 | 51.46 | **70.12** | 84.47 | **52.64** | 68.55 |
| | CIFAR-100 | 57.52 | 20.41 | 37.89 | 59.18 | **23.73** | **41.70** | 57.81 | 23.24 | 40.04 |
| $DP_{DDIM}$ | CIFAR-10 | 88.38 | 42.19 | 70.02 | 88.77 | 46.48 | **71.19** | 88.48 | **50.49** | 70.31 |
| | CIFAR-100 | 63.28 | 15.23 | 36.62 | 66.02 | 18.85 | **39.84** | 64.84 | **20.31** | 39.36 |

## M    ADDITIONAL EXPERIMENTS UNDER DETERMINISTIC WHITE-BOX SETTING

**Evaluation across different NFEs**    We also investigate the robustness under the Deterministic White-box Setting across varying NFEs. The comparison of performance between vanilla models and ADDT fine-tuned models, shown in Figure 11, highlights that ADDT consistently enhances model performance at different NFEs. This improvement is particularly pronounced at lower NFEs, further confirming that ADDT enables diffusion models to more effectively counter adversarial perturbations.

Table 22: DW-box accuracy (%) under $\ell_\infty$ perturbations for various models. ADDT consistently improves robustness across all models.

| DBP method | Vanilla | ADDT |
|---|---|---|
| $DP_{DDPM}$ | 16.80 | **39.16** |
| $DP_{DDIM}$ | 4.98 | **17.09** |
| DiffPure | 22.76 | **51.63** |
| $DP_{EDM}$ | 13.33 | **32.94** |

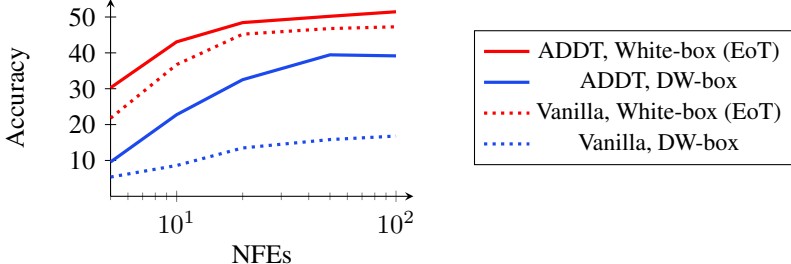

Figure 11: Revisiting Deterministic White-box Robustness. ADDT consistently improves robustness under both White-box and Deterministic White-box settings, implying that ADDT strengthens the models' ability to handle adversarial inputs.

# N    SENSITIVITY ANALYSIS OF $\lambda_{unit}$

In Section 6.1 we choose $\lambda_{unit}$=0.03 as the magnitude of such noise is similar to the adversarial perturbation of common adversarial attack setting. We also provide an ablation study here, which shows that the performance of ADDT is insensitive to $\lambda_{unit}$ and gets a consistent improvement.

Table 23: The performance of ADDT fine-tuned $DP_{DDPM}$ with varying $\lambda_{unit}$ values. ADDT shows insensitivity to $\lambda_{unit}$ and consistently improves robust accuracy (%).

| NFEs | $\ell_\infty$ | | | | $\ell_2$ | | | |
|---|---|---|---|---|---|---|---|---|
| | Vanilla | 0.02 | 0.03 | 0.04 | Vanilla | 0.02 | 0.03 | 0.04 |
| 5 | 21.78 | 24.02 | 30.27 | **31.25** | 36.13 | 41.99 | 41.99 | **49.02** |
| 10 | 36.72 | 40.92 | 43.07 | **44.92** | 55.47 | 61.72 | 62.97 | **64.45** |
| 20 | 45.21 | 48.14 | 48.44 | **50.68** | 65.23 | 67.48 | **69.82** | 69.24 |
| 50 | 46.78 | 48.83 | 50.20 | **51.07** | 68.85 | **69.82** | 69.04 | 69.53 |
| 100* | 47.27 | 48.93 | **51.46** | 50.88 | 69.34 | **70.31** | 70.12 | 69.92 |

# O    COMPUTATIONAL COST ANALYSIS FOR TRAINING AND INFERENCE

Fine-tuning DDPM and DDIM models using ADDT to achieve near-optimal performance requires 50 epochs and approximately 12 hours of training on 4 NVIDIA GeForce RTX 2080 Ti GPUs. This efficiency matches that of traditional adversarial training approaches and is notably faster than recent adversarial training techniques that utilize diffusion models for dataset augmentation (Wang et al., 2023). However, testing $DP_{DDPM}$ and $DP_{DDIM}$ involves significant computational expense due to the use of Expectation over Transformation (EoT). For instance, validating 1,024 images on the CIFAR10/CIFAR100 datasets takes approximately 5 hours on the same GPU configuration.

One of the key advantages of ADDT is its "train-once" methodology. Once the initial training is complete, ADDT can protect multiple classifiers without requiring additional fine-tuning, as demonstrated in Table 2. This is in stark contrast to adversarial classifier training, where each classifier demands individual training.

During inference, models trained with ADDT have a similar complexity to standard DBP. However, their performance gains in accelerated scenarios offer the potential for a reduction in computational overhead. As shown in Table 4, $DP_{DDPM}$ + ADDT achieves comparable performance to $DP_{DDPM}$ while requiring only 20 NFEs, resulting in up to an 80% reduction in computation time compared to the 100 NFEs required for $DP_{DDPM}$.

## P    CREDIBILITY OF OUR PAPER

The code was developed independently by two individuals and mutually verified, with consistent results achieved through independent training and testing. The open-source code is available at https://github.com/LYMDLUT/ADDT.

## Q    BROADER IMPACT AND LIMITATIONS

Our work holds significant potential for positive societal impact across a range of sectors, including autonomous driving, facial recognition payment systems, and medical assistance. We are dedicated to enhancing the safety and trustworthiness of global AI applications. However, we acknowledge the potential negative societal impacts, particularly concerning privacy protection, due to adversarial perturbations. Despite this, we believe that the overall positive impacts outweigh these risks.

In terms of limitations, our approach could benefit from incorporating insights from traditional adversarial training methods (Zhang et al., 2019; Shafahi et al., 2019; Wang et al., 2023), such as through more extensive data augmentation and a refined ADDT loss design. Nevertheless, these limitations are relatively minor and do not diminish the overall contributions of this paper. We believe these new findings and perspectives will have a sustained impact on future research in DBP, which is a promising approach to adversarial defense and could prove even more valuable for real-world applications, despite the early stage of existing studies on DBP.

