# OpenReview forum: "Towards Understanding the Robustness of Diffusion-Based Purification: A Stochastic Perspective"
_ICLR.cc/2025/Conference — ICLR 2025 Poster_

### Official Review · Reviewer_mx31 · 2024-11-01

**Soundness:** 2
**Presentation:** 3
**Contribution:** 3
**Rating:** 6
**Confidence:** 5

**Summary:**

This paper examines how stochastic elements contribute to robustness in DiffPure by systematically ablating these components (DW-box). The authors propose incorporating adversarial training into DiffPure and addressing the "robustness-generalization tradeoff" through a novel technique, RGBM, which preserves both robustness and generative performance.

**Strengths:**

I like the following contributions in this paper:

- The authors systematically study how each of the stochastic elements in DiffPure contributes to robustness, including the "forward process," "backward process," and the combination of both, clearly demonstrating how much each component contributes to robustness.

- The authors propose RGBM, which significantly alleviates the previous limitation of adversarial training compromising the generative ability of diffusion models.

- Fig. 5 is excellent. Without Fig. 5, it would be challenging to understand RGBM clearly due to existing ambiguities (see question 3 on the two types of RGBM).

**Weaknesses:**

There are no severe weaknesses in this paper, but one notable issue:

- "Note that DDPM is mathematically equivalent to the DDPM++ (VPSDE) model used in DiffPure but employs a smaller UNet architecture. DDIM, on the other hand, is a mathematical improvement over DDPM that facilitates an ODE-based diffusion process." This statement lacks formality. DDPM is a mathematical framework independent of neural network architecture. Additionally, the DDPM framework represents only one discretization of the VP-SDE, while other discretizations also exist. The authors should address this non-formal claim.

**Questions:**

I'm quite curious about the following (I would like to raise my score if the authors could provide more comprehensive studies on the mechanism of their proposed methods):

- Are the points in Fig. 1 obtained by sampling at intervals of a certain number of attack steps? If so, does this imply that the attack process initially moves the farthest distance and moves the fastest, while the loss increases the slowest, and then the movement gradually slows as time progresses but the loss increases more rapidly?

- How did you come up with RGBM? What inspired this approach?

- RGBM uses values from a Gaussian distribution but follows the order of adversarial perturbations. What would happen if we used the order of Gaussian noise but the values of adversarial perturbations? I’m curious if the main mechanism of RGBM is to eliminate large (outlier) values.

- I am very interested in whether "ADDT w/o RGBM" and "ADDT" in Table 6 were fine-tuned from pretrained models or trained from scratch. Was "ADDT w/o RGBM" fine-tuned from "ADDT"?

- Can I understand RGBM as maintaining robustness while improving generalization and generation performance across different attacks?

- I am very curious about why RGBM works. First, I notice that the result mapped by RGBM does not follow a Gaussian distribution, nor even a non-isotropic Gaussian (since each element is not Gaussian). Thus, the RGBM-mapped result is a biased estimator rather than an unbiased one. The only similarity I see to a Gaussian distribution is that, like a high-dimensional Gaussian, it also lies on a spherical shell. Is this the main reason it preserves the generalization of pretrained diffusion models? If so, simply "normalizing" the adversarial perturbation to an \(\ell_2\) norm of \(O(\sqrt{D})\) should also work (since the \(\ell_2\) norm of a high-dimensional Gaussian converges to \(\sqrt{D}\)). A simple way like yours (your illustration) to understand this is first to sample a Gaussian noise, compute its norm, and normalize the adversarial perturbation to this norm. Could you provide an ablation study on this?

---

> ### Author Response · Authors · 2024-11-22
>
> We're truly grateful for your constructive comments and appreciate the opportunity to clarify and expand on the points you have raised.
>
> > ### **W1: Clarification on the diffusion discretization forms and model architecture.**
>
> Thank you for your valuable feedback. We sincerely apologize for any confusion caused by the lack of formality in our statement.
>
> We fully agree with your points regarding diffusion frameworks. VPSDE (DDPM++) is a continuous diffusion process. It has multiple discretization forms, one of which is DDPM. Specifically, VPSDE in DiffPure adopts the same discretization form as DDPM (Appendix C in ScoreSDE [1]). To clarify, when we state "*Note that DDPM is mathematically equivalent to the DDPM++ (VPSDE) model used in DiffPure*" in line 162-163, we intend to emphasize that the mathematical differences between these frameworks are minimal and  have little influence on the defense performance.
>
> We also acknowledge your point that DDPM constitutes a mathematical framework that is distinct from any neural network architecture. In our statement that "*DDPM ... employs a smaller UNet architecture*" in line 162-163, our reference to DDPM and DDPM++ was intended to encompass both the mathematical framework and the implementation aspects. We aim to highlight that the performance differences between DP_DDPM and DiffPure are primarily due to variations in network architecture rather than the mathematical framework.
>
> We appreciate your recommendations and will address these informal expressions in our revisions.
>
> [1]Song Y, Sohl-Dickstein J, Kingma D P, et al. Score-based generative modeling through stochastic differential equations. ICLR, 2021.
>
> > ### **Q1: Clarification for the trajectory in Figure 1.**
>
> The points in Figure 1 are indeed obtained by sampling at each step of the PGD-20 attack. In this plot, the $xy$-plane represents the input space, and the $z$-axis represents the loss. At the beginning of the attack process, the points start from the same zero points (`torch.zeros_like(images)`) and are sparsely distributed along the trajectory, indicating relatively large steps and rapid increases in loss. As the attack progresses, the movements become smaller, and the points become denser, signifying finer adjustments to the perturbations.
>
> We believe this behavior is characteristic of the adversarial perturbation optimization process in PGD. During the initial steps, adversarial perturbations make larger, uniform-length movements, resulting in significant changes in the loss. After several optimization steps, the perturbations approach the attack constraints, resulting in smaller refinements in subsequent steps. While the loss continues to increase, the rate of growth slows compared to the initial stages.
>
> We hope this clarifies the dynamics depicted in Figure 1 and provides a better understanding of the attack trajectory.
>
> > ### **Q2 & Q5: The motivation and the purpose of RBGM.**
>
> In short, RBGM was inspired by the need to incorporate adversarial perturbations into diffusion model training. Traditional diffusion models rely on Gaussian noise to guide the process, and we want to emulate the Gaussian properties in the transformation of adversarial perturbations. For more details, please refer to our **General Response**.
>
> > ### **Q3: Exploring training with adversarial perturbations using Gaussian noise order.**
>
> We agree that reshaping the distribution of values is an important aspect of RBGM. As you suggest, we have run the experiment of using the order of Gaussian noise but the values of adversarial perturbations on DP_DDPM. The results are presented in the table below. Note that for a fair comparison, we normalize the adversarial perturbations by adjusting its mean and variance, since the magnitude of their original values (accumulated gradients via EoT) are much smaller than that of standard Gaussian noise.
>
> Our results indicate that using the order of the Gaussian noise with the values of the adversarial perturbations achieves lower accuracy than vanilla models on both clean and robust samples.
>
> | Accuracy(%) | Vanilla | | |Adversarial perturbations with Gaussian order |||
> | --- | --- | --- | --- | --- |--- |--- |
> | | Clean | $\ell _ {\infty}$| $\ell _ {2}$ | Clean | $\ell _ {\infty}$ | $\ell _ {2}$ |
> | 5 | **49.51** | **21.78** | **36.13** | 48.40 | 21.10 | 33.40 |
> | 10 | **73.34** | **36.72** | **55.47** | 71.78 | 34.07 | 52.98 |
> | 20 | **81.45** | **45.21** | **65.23** | 79.99 | 42.43 | 64.21 |
> | 50 | **85.54** | **46.78** | **68.85** | 83.90 | 46.73 | 68.17 |
> | 100| **85.94** | **47.27** | **69.34** | 84.54 | 46.98 | 69.33 |

---

> ### Author Response · Authors · 2024-11-22
>
> > ### **Q4: Pre-training of "ADDT w/o RGBM" and "ADDT" models.**
>
> Both "ADDT w/o RGBM" and "ADDT" models are fine-tuned from the same pre-trained diffusion models. We will clarify this point better in the paper.
>
> > ### **Q6: Experiments for adversarial perturbation with $\ell _ {2}$ normalization.**
>
> We acknowledge that mapping perturbations to a spherical shell is a beneficial aspect of RBGM. However, we contend that RBGM encompasses more than this, as discussed in our **General Response**.
>
> We compare DP_DDPM adopting models trained with $\ell _ {2}$-normalized perturbations and those with RBGM-mapped perturbations as you suggest. The results in the table below reveal that although models trained with $\ell _ {2}$-normalized perturbations perform better in some circumstances under $\ell _ {2}$ attacks (potentially because these perturbations are more similar to the perturbations produced by $\ell _ {2}$ attack during testing), ADDT exhibits consistently higher robustness under $\ell_{\infty}$ attacks. Furthermore, ADDT achieves comparable performance under $\ell_{2}$ attacks and achieves higher performance in scenes with more NFEs. In addition, ADDT exhibits a lower FID value, indicating better preservation of generative capability.
>
> | Accuracy (%) | Perturbation with $\ell _ {2}$ normalization | | | ADDT (w/ RBGM) |||
> | --- | --- | --- | --- | --- | --- | --- |
> | NFEs | Clean | $\ell _ {\infty}$ | $\ell _ {2}$ | Clean | $\ell _ {\infty}$ | $\ell _ {2}$ |
> | 5 | **60.40** | 28.47 | **44.58** | 59.96 | **30.27** | 41.99 |
> | 10 | **79.29** | 41.90 | **63.72** | 78.91 | **43.07** | 62.97 |
> | 20 | 83.59 | 47.85 | 67.68 | **83.89** | **48.44** | **69.82** |
> | 50 | 84.83 | 49.12 | **69.24** | **85.45** | **50.20** | 69.04 |
> | 100 | 84.97 | 49.95 | 69.29 | **85.64** | **51.46** | **70.12** |
>
> | | Clean fine-tuning   | ADDT (w/ RBGM) | Perturbations with $\ell _ {2}$ normalization |  ADDT w/o RBGM   |
> | --- | --- | --- | --- | --- |
> | FID | 3.50 |  5.190   | 5.678 |  13.608  |
>
> To explain these results, we propose that the conventional training of diffusion models is predicated on the assumption:
>
> $x _ t = \sqrt{\overline{\alpha} _ t} x _ 0 + \sqrt{1 - \overline{\alpha} _ t} \epsilon$.
>
> whereas ADDT can be represented as:
>
> $x _ t = \sqrt{\overline{\alpha} _ t} x_0 + \sqrt{1 - \lambda _ t^2} \sqrt{1 - \overline{\alpha} _ t} \epsilon + \lambda _ t \sqrt{1 - \overline{\alpha} _ t} \epsilon _ {\delta}(\delta)$.
>
> During ADDT training, our goal is to design perturbations that mimic the characteristics of traditional diffusion models. Both RBGM and $\ell _ {2}$ normalization should be considered as certain approximations in this context. As discussed in our **General Response**, RBGM may emerge as a more accurate approximation.

---

> ### Author Response · Authors · 2024-11-30
>
> Dear Reviewer mx31:
>
> We sincerely appreciate your thorough and constructive review. Your detailed feedback has been invaluable in helping us refine and strengthen our research paper.
>
> In response to your feedback, we have provided detailed explanations and made revisions to our paper. Should you have any further questions or need additional clarification, please do not hesitate to discuss with us.

---

### Official Review · Reviewer_nzsE · 2024-11-03

**Soundness:** 3
**Presentation:** 3
**Contribution:** 3
**Rating:** 8
**Confidence:** 3

**Summary:**

The paper explores the reason behind the success of Diffusion-Based Purification (DBP) as a defense against adversarial attacks, highlighting that its effectiveness comes from randomness rather than just reducing the difference between clean and adversarial images. The authors introduced a new evaluation method, Deterministic White-Box (DW-box), and found that DBP’s ability to purify adversarial perturbations is weak without stochasticity. To improve DBP, they developed Adversarial Denoising Diffusion Training (ADDT), which incorporates adversarial examples into the training process, and Rank-Based Gaussian Mapping (RBGM), which makes adversarial perturbations more Gaussian-like. Their experiments showed that these methods enhance DBP’s robustness.

**Strengths:**

1) The paper is well written and relatively easy to follow.
2) Novel perspective on why Diffusion-Based Purification works and how it can be further improved.
3) Claims are well supported by experiments.

**Weaknesses:**

N/A

**Questions:**

1) Based on the conclusion of your paper that stochasticity is the main ingredient of the success of the DBP, do you think a diffusion model even necessary for the defense? What if we just add a certain amount of gaussian noise directly to the adversarial example and then just denoise it with an off-the-shelf image denoising network that was specifically trained to remove gaussian noise? Do you think it will work just as well as a defense method?

---

> ### Author Response · Authors · 2024-11-22
>
> > ### **Q1: Whether diffusion models are necessary for stochasticity-based adversarial defense.**
>
> Thank you for your insightful question.
>
> We appreciate your interest in exploring alternative stochasticity-based defense methods without using diffusion models. Indeed, DBP is not the only randomized defense method, and several different approaches exist, such as denoising autoencoders [1] and randomized masking [2]. However, among these approaches, DBP stands out by demonstrating better robustness against adaptive attacks.
>
> As you suggest, we explore whether using an off-the-shelf image denoising network without an iterative reverse process can also produce relatively good defensive performance. Here we introduce a consistency model [3] which only requires one denoising step for each inference. It uses the same UNet as DiffPure and is distilled from a diffusion model. We set the forward noise strength to 0.5, which is higher than DiffPure for more stochasticity. It could be represented as
>
> $x _ \text{purified} = \text{UNet}(x _ 0 + 0.5 \times \epsilon)$.
>
> We apply this purification method to the same pre-trained WRN-28 classifier as in our paper, and it achieves 49.32% under $\ell _ {\infty}$ attack with PGD20+EoT10, as shown in the table below. However, its performance still lags behind DiffPure. This can be explained by the stronger stochasticity of DiffPure due to its iterative random reverse processes, as indicated by the comparison of DDPM and DDIM in Sec. 4.1.
>
> We would like to further emphasize that not all image denoising networks work well. As an example, we keep the noise strength and the UNet architecture identical and only change the training method of the denoising network. Specifically, we study two models: (1) a denoising UNet trained with $\ell _ {2}$ reconstruction loss, and (2) a UNet jointly trained with the classifier using cross-entropy loss. The results in the table below suggest that applying these two models for defense can also provide non-trivial robust accuracy, but the performance is significantly lower than using diffusion or consistency models.
>
> Given these preliminary results, we believe that the question you raised is valuable and worthy of further exploration in the future.
>
> | PGD20+EoT10(%) | Clean | $\ell _ {\infty}$ | $\ell _ {2}$ |
> | --- | --- | --- | --- |
> | DiffPure | 89.26 | 55.96 | 75.78 |
> | Consistency model | 83.69 | 49.32 | 67.58 |
> | UNet with $\ell _ {2}$ loss | 74.22 | 35.26 | 54.32 |
> | UNet+cls with CE loss | 77.02 | 24.34 | 56.18 |
>
>
> [1] Gu S, Rigazio L. Towards deep neural network architectures robust to adversarial examples. ICLR, 2015.
> [2] Ughini G, Samele S, Matteucci M. Trust-no-pixel: A remarkably simple defense against adversarial attacks based on massive inpainting. IJCNN, 2022.
> [3] Song Y, Dhariwal P, Chen M, et al. Consistency models. ICML, 2023.

---

> > ### Comment · Reviewer_nzsE · 2024-11-27
> >
> > Thank you for your response. I will keep my score.

---

> ### Author Response · Authors · 2024-11-28
>
> Dear Reviewer nzsE:
>
> Thank you for the time and expertise you've dedicated to reviewing our work. We are deeply grateful for your continued support. Your encouragement and feedback are invaluable.

---

### Official Review · Reviewer_gEWD · 2024-11-03

**Soundness:** 3
**Presentation:** 2
**Contribution:** 3
**Rating:** 6
**Confidence:** 4

**Summary:**

This paper investigates the robustness of Diffusion-Based Purification (DBP) methods in adversarial attacks, proposing that its defensive effectiveness relies primarily on stochasticity rather than the traditional noise ablation process. By introducing a deterministic white-box evaluation approach, the authors demonstrate the critical role of stochasticity in DBP's robustness. To enhance DBP's denoising capability, the paper presents Adversarial Denoising Diffusion Training (ADDT) and Rank-Based Gaussian Mapping (RBGM), with experimental results showing significant improvements in defense performance. The authors suggest that future research should focus on decoupling stochasticity from purification capability to optimize defensive performance.

**Strengths:**

1. The paper identifies that the robustness provided by diffusion models is likely due to stochastic elements within the model rather than solely the diffusion process.
2. The authors propose a new training method for diffusion models that incorporates robustness information, enhancing the model's purification capability.

**Weaknesses:**

1. While the study contributes positively to understanding purification using diffusion models, there are concerns about the complexity of purification. It significantly increases the model's complexity, potentially multiple times compared to the original model, and this real-world cost needs to be experimentally clarified (complexity analysis).
2. If the training data for the diffusion model originates from the same data as the original model, a comparison with methods using the diffusion model to generate new samples for robust model training would be valuable, as this approach does not add runtime complexity [1].
3. Questions remain about why perturbations are not directly modified by mean and variance adjustments to achieve Gaussian distribution, instead requiring replacement.

**Reference**:

[1] Wang, Zekai, et al. "Better diffusion models further improve adversarial training." International Conference on Machine Learning. PMLR, 2023.

**Questions:**

Please see Weaknesses.

---

> ### Author Response · Authors · 2024-11-22
>
> Thank you for your constructive feedback, which provided us with valuable directions for revising our paper.
>
> > ### **W1: Complexity analysis and comparison.**
>
> Prior work [1] provides a discussion on the inference costs associated with Diffusion-Based Purification (DBP) methods. We cite some of the results in the table below, which shows the inference time (seconds), with the increase of $t^* = 0.1$ over $t^* = 0$ indicated in parentheses.
>
> | Dataset | Network | $t^* = 0$ | $t^* = 0.1$ |
> | --- | --- | --- | --- |
> | CIFAR-10 | WRN-28-10 | 0.055 | 10.56 ($\times$ 190) |
> | ImageNet | ResNet-50 | 0.062 | 11.13 ($\times$ 179) |
>
>
> While sharing a complexity profile with standard DBP, our method significantly reduces computational overhead through the use of ADDT. As shown in Table 3 in our paper, with just 20 NFEs, we achieve performance on par with DP_DDPM at 100 NFEs, reducing computational time by up to 80%.
>
>
>
> > ### **W2: Comparison with AT adopting diffusion models for data augmentation.**
>
> We compare the robust accuracy of DBP models and AT models that adopt diffusion models for data augmentation [2]. The evaluation was conducted on WRN-28-10 under the PGD20 attack (with PGD20+EoT10 for DiffPure and ADDT). The results are summarized in the table below.
>
> |  |  | Clean | $\ell _ {\infty}$ | $\ell _ {2}$ |
> | :---: | :---: | :---: | :---: | :---: |
> | AT | $\ell _ {\infty}$ model | 93.26 | 72.07 | 74.71 |
> | | $\ell _ {2}$ model | 94.92 | 56.93 | 83.98 |
> | AP | DiffPure | 89.26 | 55.96 | 75.78 |
> | | DiffPure+ADDT | 89.94 | 62.11 | 76.66 |
>
>
> While AT performs better under the same norm as training, ADDT shows improved performance on adversarial examples under different norms, suggesting better generalization against unseen attacks.
>
> > ### **W1&W2: Broader discussion of DBP and AT adopting diffusion models for data augmentation.**
>
>
> DBP and AT methods fundamentally differ in their mechanisms and applications. Mechanistically, DBP leverages stochasticity to avoid the most effective attack directions. In contrast, diffusion models adopted by AT enhance robustness indirectly by generating synthetic images to expand the training dataset. From the perspective of computation cost, DBP does not alter the training of the classifier, and a single diffusion model can be used to protect multiple classifiers, offering adaptability across various tasks. However, it incurs higher inference costs due to its iterative purification process. In contrast, AT adopting diffusion models for data augmentation requires independent and resource-intensive training for each classifier, while no additional inference cost is needed.
>
> In this paper, the effectiveness of ADDT highlights the potential synergy between DBP and established AT methods for further performance enhancement. It is also important to note that DBP is still in its early stages of exploration, and techniques such as model distillation [3][4] and latent space diffusion[5] hold great potential for further accelerating DBP. Hence, we believe that further studies on DBP are worthwhile.
>
> > ### **W3: Why replacement is required for RBGM.**
>
> While it is possible to adjust the distribution of perturbation values to align their mean and variance, this does not ensure that the overall distribution of the values exhibits Gaussian characteristics. Specifically, even if the mean and variance are aligned, the underlying distribution of perturbation values can vary significantly, e.g., some may be more clustered, while others might display a broader and more uniform spread. In addition, the distribution of perturbation values across different images and timesteps $t$ may have various characteristics, as illustrated in our **General Response**.
>
> By adopting RBGM, we ensure that the values of these perturbations consistently follow a Gaussian distribution, regardless of the images or timesteps. Meanwhile, it brings stochasticity and reduces the risk of overfitting specific images. More details can be found in our **General Response**.
>
> We hope this response addresses your concerns.
>
> [1] Nie, Weili, et al. Diffusion Models for Adversarial Purification. ICML, 2022.
> [2] Wang, Zekai, et al. Better diffusion models further improve adversarial training. ICML, 2023.
> [3] Song Y, Dhariwal P, Chen M, et al. Consistency models. ICML, 2023.
> [4] Salimans T, Ho J. Progressive Distillation for Fast Sampling of Diffusion Models. ICLR, 2022.
> [5] Rombach R, Blattmann A, Lorenz D, et al. High-resolution image synthesis with latent diffusion models. CVPR, 2022.

---

> ### Author Response · Authors · 2024-11-30
>
> Dear Reviewer gEWD:
>
> We sincerely appreciate your thorough and constructive review. Your detailed feedback has been invaluable in helping us refine and strengthen our paper.
>
> In response to your feedback, we have provided detailed explanations and made revisions to our paper. Should you have any further questions or need additional clarification, please do not hesitate to discuss with us.

---

> > ### Comment · Reviewer_gEWD · 2024-12-02
> >
> > Thank you for your response. It has addressed my concerns, and I will maintain my score.

---

> > > ### Author Response · Authors · 2024-12-02
> > >
> > > Dear Reviewer gEWD:
> > >
> > > Thank you for your thorough review of our paper. We greatly appreciate the time and effort you have dedicated. Your insightful feedback and continued support are invaluable to us.

---

### Official Review · Reviewer_TLr9 · 2024-11-04

**Soundness:** 4
**Presentation:** 3
**Contribution:** 4
**Rating:** 6
**Confidence:** 4

**Summary:**

The author argues that the current effectiveness of the DBP method, attributed to the forward process of diffusion models, lacks sufficient empirical validation. While they acknowledge the theory of mitigating distribution gap, they question whether the forward and backward diffusion stochasticity in DBP is indeed what enhances model robustness. To address this, they propose a new attack evaluation, the DW-Box attack, which captures both stochasticity information and parameters of defense system. Experiments reveal that models like DDPM and DDIM lack strong robustness against this new attack. Additionally, the authors introduce ADDT, a fine-tuning technique for diffusion models in the existing DBP methods. This approach first optimizes the generated adversarial perturbation using classifier guidance, and then performs further training on images with both Gaussian noise and optimized perturbations. They aim to enhance the purification capability of diffusion models. This method significantly enhances the robustness of existing DBP methods. Overall, the paper critically examines the underlying principles that make DBP methods effective.

**Strengths:**

1. The work has included extensive experiment contents, such as exploration on accelerated generation of diffusion model, stronger white-box attack with more attack iterations and included various dataset results.

2. The author proposed an appropriate attack evaluation, DW-box attack, which makes the performance comparison reliable.

3. The ADDT technique has improved the robustness of all baseline DBP methods.

**Weaknesses:**

1. The author should provide a few more comparisons for the same content in figure 6, which shows the performance of the ADDT fine-tuning under the proposed DW-box attack.

2. In figure 3, it is not convincing if the author only draws one complete attack trajectory.

3. The whole idea can be viewed as an additional training by using samples with more Gaussian noises at the same timestep, hence the diffusion model achieves better purification ability, I would like to see whether the RBGM perturbation is different with using a pure additional Gaussian noise with trivial order. Hence, we cannot say the model has improved its ability of enhancing purification of adversarial perturbations.

**Questions:**

1. As the perturbation after Rand-Based Gaussian Mapping is still a Gaussian noise, can I view the idea of ADDT is just using further perturbed images for additional training, if so, is the improvement relying on the additional data?

2. What is the reason of choosing such a range for the modulation term in line 362-364?

---

> ### Author Response · Authors · 2024-11-22
>
> We are grateful for your detailed feedback, which is invaluable in improving the clarity and depth of our paper.
>
> > ### **W1：More results under DW-box attack.**
>
> We perform further experiments on the VPSDE and EDM models under the proposed Deterministic White-Box (DW-box) attack. These additional results show that ADDT provides a consistent improvement.
>
> | DW-box accuracy under $\ell _ \infty$ (%) | Vanilla | ADDT |
> | --- | --- | --- |
> | DP_DDPM | 16.80 | **39.16** |
> | DP_DDIM | 4.98 | **17.09** |
> | **DiffPure (VPSDE)** | 22.76 | **51.63** |
> | **DP_EDM (EDM)** | 13.33 | **32.94** |
>
> Additionally, we have also provided a comparison between Vanilla and ADDT models under different NFEs for DP_DDPM in White-box and DW-box attacks in Figure 11 of Appendix N. These results also demonstrate the consistent improvement.
> > ### **W2: Clarification on Figure 3.**
>
> We apologize for any misunderstanding caused by the initial representation. The trajectory depicted in Figure 3 actually averages over 128 images, as mentioned in the caption and detailed in Appendix F. To create this plot, we project the attack trajectories onto the plane defined by the deterministic white-box attack direction and the orthogonal direction. We then average the attack trajectories on 128 images to obtain a clear and reliable representation.
>
> > ### **W3 & Q1: The differences between RBGM-mapped perturbations and pure Gaussian noise.**
>
> Intuitively speaking, the RBGM-mapped perturbation could be seen as "picked from a Gaussian distribution"; however, **its distribution is no longer purely Gaussian** and is "more adversarial" than Gaussian noise. Hence, ADDT is more akin to adversarial training (AT) rather than simply training with additional data. Further discussions on RBGM can be found in the **General Response**.
>
> Here we provide an additional experiment justifying the difference between RBGM-mapped perturbations and Gaussian noise. We mix clean images with 0.03 times Gaussian noise and RBGM-mapped perturbations. The results are as follows, showing that RBGM-mapped perturbations are indeed adversarial to the model.
>
> | Accuracy (%) | Clean image | Clean image + Gaussian noise | Clean image + RBGM-mapped perturbation |
> | --- | --- | --- | --- |
> | WRN-28-10 | 95.12 | 81.54 | 4.47 |
>
>
> Hence, through ADDT and RBGM, we integrate adversarial perturbations into the training of DBP models, equipping them with the power to directly counter adversarial perturbations. We hope this clarification resolves your confusion, and we will also provide clearer explanations in the final revision.
>
>
> > ### **Q2: Reason for choosing the range of the modulation term in Lines 362-264.**
>
> In traditional adversarial attacks, perturbations are directly added to the image, resulting in an adversarially modified image $x_{adv} = x_0 + \delta$, where $x_0$ denotes the original input image, and $\delta$ represents the adversarial perturbation. In our approach ADDT, we integrate this concept into the diffusion process, yielding $x_t = \sqrt{\overline{\alpha}_t} (x_0 + \delta) + \sqrt{1 - \overline{\alpha}_t} \epsilon$. To train a diffusion model capable of purifying such perturbations, we combine the perturbation and noise, reformulating the equation as: $x_t = \sqrt{\overline{\alpha} _ t} x_0 + \sqrt{1 - \overline{\alpha} _ t} (\epsilon + \frac{\sqrt{\overline{\alpha} _ t}}{\sqrt{1 - \overline{\alpha} _ t}} \delta)$, where the term $\frac{\sqrt{\overline{\alpha} _ t}}{\sqrt{1 - \overline{\alpha} _ t}}$ is denoted as $\gamma _ t$. Given that $\alpha _ t$ ranges from 0 to 1, $\gamma _ t$ spans from 0 to $\infty$. To ensure the adversarial perturbation remains within a manageable intensity, we constrain its value to lie between $\lambda _ {\text{min}}$ and $\lambda _ {\text{max}}$.
>
>
> The selection of the modulation term $\lambda_{\text{unit}}$ is based on our experimental findings. As shown in Table 23 in Appendix O, the performance of ADDT is relatively insensitive to the specific value of $\lambda _ {\text{unit}}$ within the tested range, showing consistent improvement across different attack settings. We choose $\lambda _ {\text{unit}} = 0.03$ as it achieves generally better performance on both $\ell _ \infty$ and $\ell _ 2$ attacks.
>
> Thank you again for your thorough and thoughtful feedback. We will improve the paper according to our replies above.

---

> > ### Comment · Reviewer_TLr9 · 2024-11-26
> >
> > Thank you for your response. I will continue to support the acceptance of this paper. Best of luck to you!

---

> ### Author Response · Authors · 2024-11-28
>
> Dear Reviewer TLr9:
>
> Thank you for your support. We are deeply grateful for your positive feedback on our paper. Your constructive comments have been invaluable in refining our work, and we have carefully addressed them in the revised version.
>
> We sincerely appreciate your recognition of our efforts and your encouragement throughout the discussion.

---

### Comment · Area_Chair_ccqy · 2024-11-22

Dear Authors and Reviewers,

The discussion phase has passed 10 days. If you want to discuss this with each other, please post your thoughts by adding official comments.

Thanks for your efforts and contributions to ICLR 2025.

Best regards,

Your Area Chair

---

### Author Response · Authors · 2024-11-22
**General Response Regarding RBGM**

**Motivation of RBGM.**

In Section 4, we discuss the limitations of Diffusion-Based Perturbation (DBP) models in directly countering adversarial perturbations. To overcome these limitations and simultaneously preserve the generative ability of the diffusion models, we introduce a novel approach: incorporating "adversarially selected Gaussian noise" into the training process.

To elaborate, a conventional diffusion forward process is based on the equation:

$x _ t = \sqrt{\overline{\alpha} _ t} x _ 0 + \sqrt{1 - \overline{\alpha} _ t} \epsilon,$

where $x_t$ represents the noisy image at time $t$, $x_0$ is the initial input, $\overline{\alpha}_t$ is a time-dependent scaling factor, and $\epsilon$ is random Gaussian noise. Our proposed method, ADDT, modifies this equation to include an adversarial component:

$x _ t = \sqrt{\overline{\alpha} _ t} x_0 + \sqrt{1 - \lambda _ t^2} \sqrt{1 - \overline{\alpha} _ t} \epsilon + \lambda _ t \sqrt{1 - \overline{\alpha} _ t} \epsilon _ {\delta}(\delta).$

In this revised formulation, $\epsilon _ {\delta}(\delta)$ represents the adversarial perturbation, and $\lambda_t$ is a parameter that controls the blend between traditional and adversarial noise. The core objective of ADDT training is to ***generate perturbations that emulate the characteristics of Gaussian noise in conventional diffusion training while incorporating adversarial disturbances***.

This introduces our Rank-Based Gaussian Mapping (RBGM) technique, which retains the relative ordering of perturbation magnitudes while adjusting the values to more closely resemble a Gaussian distribution. **The importance of RBGM is twofold**:

**(1) Enhancing statistical consistency.**

Raw adversarial perturbation values often exhibit non-standard distributions, and RBGM serves to recalibrate these perturbations, aligning them more closely with a Gaussian distribution. To elaborate, rather than enforcing a multivariate Gaussian distribution for the entire perturbation, RBGM ensures that the distribution of individual perturbation values adheres to Gaussian characteristics.


The benefit of this transformation can be illustrated in the following figures. For a fair comparison, the perturbation values have been normalized. In the first figure, the original perturbation values display a wide array of distributions across different images and timesteps. After the mapping of RBGM, these values are transformed to exhibit a uniform Gaussian distribution.

https://anonymous.4open.science/r/ICLR_ADDT/distribution_of_perturbation_values.pdf

In the second figure, the raw perturbations show irregular and inconsistent behavior when mixed with Gaussian noise at varying ratios. However, after RBGM adjustment, the perturbations and the mixture exhibit consistent statistics with the pure Gaussian noise. The statistical consistency of the perturbation values may ease the training of the diffusion model and avoid significant deviation from the normal diffusion process.

https://anonymous.4open.science/r/ICLR_ADDT/visual_compare_ability_of_RBGM.pdf

**(2) Reducing image-specific dependence.**

In the training of diffusion models, the Gaussian noise is independent of specific images or timesteps. This approach contrasts with the nature of adversarial perturbations, which are typically tailored to each input. RBGM mitigates this by introducing stochasticity into the construction of perturbations and merely preserving the ranks of values of the image-dependent adversarial perturbations, thus reducing image-specific dependence. This characteristic further ensures the resemblance of ADDT to the diffusion training process and potentially mitigates the over-fitting of training images.

---

### Meta-Review · Area_Chair_ccqy · 2024-12-21

**Metareview:**

This paper focuses on understanding diffusion-based purification which is an effective defense mechanism against adversarial attacks in recent literature. The paper identifies that the robustness provided by diffusion models is likely due to stochastic elements within the model rather than solely the diffusion process. The authors propose a new training method for diffusion models that incorporates robustness information, enhancing the model's purification capability. The work has also included extensive experiment contents, such as exploration on accelerated generation of diffusion model, stronger white-box attack with more attack iterations and included various dataset results. During rebuttal, all concerns are addressed by the authors. Due to the significance of this paper, I recommend accepting this paper at this round.

**Additional Comments On Reviewer Discussion:**

All concerns are well-addressed by the authors during the rebuttal.

---

### Decision · Program_Chairs · 2025-01-22

Accept (Poster)